# Changes in contraceptive and sexual behaviours among unmarried young people in Nigeria: Evidence from nationally representative surveys

Sunday A. Adedini [1,2]*, Jacob Wale Mobolaji[3], Matthew Alabi[3], Adesegun O. Fatusi[4,5,6]

**1** Demography and Social Statistics Department, Faculty of Social Sciences, Federal University Oye-Ekiti, Oye-Ekiti, Nigeria, **2** Programme in Demography and Population Studies, Schools of Public Health and Social Sciences, University of the Witwatersrand, Johannesburg, South Africa, **3** Department of Demography and Social Statistics, Obafemi Awolowo University, Ile-Ife, Nigeria, **4** Academy for Health Development (AHEAD), Ile-Ife, Nigeria, **5** University of Medical Sciences, Ondo, Ondo State, Nigeria, **6** Department of Community Health, College of Health Sciences, Obafemi Awolowo University, Ile-Ife, Nigeria

* sunday.adedini@gmail.com

**Data Availability Statement:** Data are from the DHS Program [https://dhsprogram.com/data/]. Data from the following country were used:

## Abstract

### Context

Nigeria is a high-burden country in terms of young people's health. Understanding changes in young people's sexual and reproductive health (SRH) behaviours and the associated factors is important for framing appropriate interventions.

### Objective

This study assessed changes in SRH behaviours of unmarried young people aged 15–24 and associated factors over a ten-year period in Nigeria.

### Data and method

We analysed datasets from Nigeria Demographic and Health Surveys of 2008, 2013 and 2018 to assess changes in inconsistent condom use, non-use of modern contraceptives; multiple sexual partnership; and early sexual debut. Using binary logistic regression, we assessed the association of selected variables with the SRH behaviours.

### Results

Over four-fifths of unmarried young people (15–24) in Nigeria engaged in at least one risky sexual behaviour in each survey year. The pattern of changes in the four risky SRH behaviours was consistent over the 10-year period, with the highest rates of each behaviour occurring in 2018 while the lowest rates were in 2013, thus indicating an increase in the proportion of respondents engaging in risky sexual behaviours over the study period. Comprehensive HIV/AIDS knowledge, male gender, older age category (20–24), residence in south-west Nigeria, urban residence, higher socio-economic status, secondary/higher education were mostly protective against the four SRH variables analysed across the different data waves.

Nigeria. The authors did not have special access privileges.

**Funding:** This work was partially supported by the Consortium for Advanced Research Training in Africa (CARTA), Nairobi, Kenya. SAA and JWM are recipients of the CARTA fellowship. The funders had no role in study design, data collection and analysis, decision to publish, or preparation of the manuscript. There was no additional external funding received for this study.

**Competing interests:** The authors have declared that no competing interests exist.

## Conclusion

Addressing the high and increasing level of risky SRH behaviours among young people in Nigeria is imperative to improve overall national health status and to ensure progress towards achieving SDG target 3.7 focusing on SRH.

## Introduction

Young people (aged 10–24 years) in Nigeria are a high risk and vulnerable group for poor sexual and reproductive health (SRH) outcomes such as early and unplanned pregnancy, unsafe abortion, and sexually transmitted infections (STIs), including HIV/AIDS. The Lancet Commission on Adolescent Health and Wellbeing classifies Nigeria as a high-burden country for adolescent and youth health problems with SRH as a significant contributor [1]. Among others, Nigeria has one of the highest burdens of Human Immunodeficiency Virus (HIV) in the world and young people aged 15–24 years are disproportionately affected by the infection, and contribute about a third of all the country's new HIV infections [2, 3]. Adolescents and young people (AYP) also contribute significantly to Nigeria's high maternal mortality burden–the second highest among all countries [4], as about a fifth of adolescent girls in Nigeria have commenced childbearing in 2018 [5] and have higher risks of maternal complications compared to older age groups [6, 7]. Unsafe abortion contributes significantly to Nigeria's high maternal mortality burden. The country's abortion rate of 36 per 1000 women aged 15–49 years is higher than the sub-Saharan African's average of 28 per 1000 [8, 9], and young females contribute significantly to this due to high rates of unsafe abortion and complications with unplanned pregnancy as a driving factor [10].

Unprotected sexual intercourse is the immediate common underlying behavioural factor for HIV and other sexually transmitted diseases as well as unplanned pregnancy. Several studies have documented low rates of use of condom as well as other forms of modern contraceptives among young people in Nigeria [11–13]. Other risky sexual behaviours such as early sexual debut and multiple sexual partnership also contribute significantly to poor SRH outcomes. Several factors are associated with adolescent low contraceptive behaviour, including demand-side factors such as poor health-seeking behaviour, supply-side factors such as low availability of adolescent-and-youth-responsive health services and health workers' bias, and structural factors such as social norms and socioeconomic differentials [11, 14–16]. Scholars have adduced that unmarried adolescents are at higher risk of having unmet need for contraception than the older population due to social pressure, contraceptive access barriers arising from providers' bias, and gender-based violence among others [17–19], and the adverse consequences of unmet need among young people such as unintended pregnancies, maternal mortality, and sexually transmitted infections, including HIV/AIDS.

A review of SRH behaviour among young people in sub-Saharan Africa reflects an interesting contrast between males and females [20]. While higher economic status was a protective factor against early sexual initiation among young women, a reverse situation was the case among young men. Also, females mostly reported health care workers as their main source of contraceptive information while males mainly reported the media, friends and internet [21].

The literature shows considerable regional variations in sexual behaviours among young people in Nigeria, for example in the timing of first sex among young people in northern and southern regions, indicating the significance of spatial and cultural contexts in SRH behaviour [22]. Evidence is, however, mixed on the influence of socio-economic characteristics on sexual

behaviour of young people. A Nigerian study established early sexual initiation among educated youths [22], while education was found to be protective against early sexual debut among young people in other studies [11, 15, 16].

As part of the efforts to address HIV, poor maternal health and young people's health and developmental challenges, Nigeria has witnessed several adolescent and youth SRH policy and programme interventions over the last two decades, many of which have been largely ineffective. Assessing changes in young people's contraceptive and sexual behaviour is important for gauging the success of previous intervention efforts, monitoring progress on the target 3.7 of the Sustainable Development Goals (which aims to ensure universal access to sexual and reproductive health-care services by 2030)., as well as for prioritising future actions. Therefore, this study examined changes in contraceptive and sexual behaviour of young unmarried men and women (a highly vulnerable group in terms of reproductive health risks) in Nigeria over a ten-year period (2008–2018).

## Data and methods

### Data source

This study utilized the three most recent datasets from the Nigeria Demographic and Health Surveys (NDHS)–a repeated cross-sectional survey–conducted over a ten-year period (2008, 2013 and 2018). NDHS is a nationally representative survey that collects key demographic and health information on men aged 15–59 and women of reproductive ages (15–49 years). The three surveys were conducted using the sampling frame and clusters (primary sampling units) defined based on enumeration areas (EAs) designed for 2006 population and housing census [5, 23]. Nationally representative samples of 36,800, 40,680 and 42,000 households were randomly selected for 2008, 2013 and 2018 NDHS respectively. Respondents for the surveys were sampled using a two-stage stratified cluster sampling design. Detailed information on the sampling design and data collection strategies have been reported elsewhere [5, 23, 24].

This study focused on the never-married but sexually active young men and women aged 15–24. For the purpose of this study, the sexually active were the young men and women who reported to have ever engaged in sexual intercourse. The purpose of this operational definition was to identify the maximum number of young men and women who had experienced sexual encounter. We examined changes in contraceptive and sexual behaviours of the young men and women over 10 years (2008–2018) based on the understanding that a ten-year period is a reasonably long period of time during which SRH behaviours of young people could respond to changes in structural contexts and programmatic interventions. Analytic weighted samples for the three surveys, focusing on age 15–24, were 1,618 men and 2,574 women for 2008 NDHS; 1,702 men and 2,461 women for 2013 NDHS; and, 671 men and 2,321 women for 2018 NDHS. The NDHS uses standardized methodologies across time and space; hence key questions relating to the variables used in our analysis were framed in the same way across all the three surveys.

### Variables measurements

Our analysis focussed on four key SRH risk behaviours as outcome variables based on extant literature: (i) non-use of any modern contraceptive (ii) inconsistent condom use (iii) multiple sexual partnership and (iv)sexual debut before age 18. The variables were derived from participants' responses to the relevant questions on contraceptive use and sexual behaviours. For non-use of any modern contraceptive, sexually active respondents were asked: "which method are you or your partner currently using to delay or avoid getting pregnant?" Respondents were categorised as users of modern contraceptive if they indicated the use of any modern methods

such as oral pill, intrauterine device (IUD), injectable contraceptives, male condom, male sterilization, female condom, or female sterilization. Respondents who reported not using any contraceptive method or using non-modern contraceptives such as periodic abstinence, withdrawal and standard day methods were categorised as non-users of modern contraceptives. For condom use, sexually active respondents were asked: "was a condom used every time you had sexual intercourse with the last or second-to-the-last or third-to-the-last sexual partner in the last 12 months?" Respondents who indicated using a condom in sexual engagement with their last four sexual partners in the last 12 months or all the partners they had, if less than four, within the defined period were categorized as consistently using condom, while those not meeting this criterion were regarded as not using condom consistently. Multiple sexual partnership was defined based on the number of persons with whom the respondents have had sexual intercourse in the 12 months before the survey: individuals with at least two sexual partners were defined as having multiple partners. Age at first sexual intercourse was defined based on the report of the sexually experienced respondents as to when they ever had first sexual intercourse–either forced or voluntary. Respondents who had the first sexual intercourse before age 18 were grouped as having sexual debut before age 18.

Guided by extant literature, we explored key socio-demographic variables (sex, education, socio-economic level (measured by wealth quintiles) rural-urban location, and regions), comprehensive knowledge of HIV, and media exposure as explanatory variables for risky SRH behaviour. Table 1 presents the operational definitions for the explanatory variables.

## Statistical analysis

For univariate analysis, we assessed the percentage distribution of young people (age 15–24) by selected socio-demographic characteristics. We undertook bivariate analysis–Chi-square test–to assess the association between each of the selected explanatory variables and the

**Table 1. Explanatory variables for modelling contraceptive use and sexual behaviour among young people in Nigeria.**

| Key variables | Operational Definition | Grouping for analytical purpose |
|---|---|---|
| Sex | Sex of the respondent | (1) Male, (2) Female |
| Age | Self-reported age of respondent at time of survey | (1) 15–19 (2) 20–24 |
| Education | Highest level of education attained by respondent: | (1) None/Primary |
| | | (2) Secondary |
| | | (3) Tertiary |
| Religion | Religious affiliation of the respondents | (1) Christianity |
| | | (2) Islam |
| | | (3) Traditional and others |
| Wealth quintile | Wealth quintiles was generated in NDHS as a measure of economic status based on possession of specific household items | (1) poorest, (2) poorer, (3) middle, (4) richer, (5) richest |
| Residence | Place of residence where respondent lived at the time of survey | : (1) Urban (2) Rural |
| Region | Geo-political zones of residence where respondent lived at the time of survey | (1) North-central (2) North-east (3) North-west (4) South-east (5) South-south (6) South-west |
| Media exposure | Access to information through media: | (1) No media access (2) Had media access |
| Comprehensive HIV knowledge | Based on combination of: | (1) No comprehensive knowledge |
| | (a) lack no knowledge of all the five primary prevention methods primary prevention methods | |
| | (b) Has knowledge of all the five primary prevention methods namely: use of condoms and having just one uninfected faithful that reduce the chance of getting HIV; Knowing that a healthy-looking person can have HIV; and Rejection the two most common local misconceptions about HIV/AIDS transmission or prevention. | (2) Had comprehensive knowledge |

outcome measures. In the multivariable analysis, we assessed the effect of the selected explanatory variables on the outcome variables, using binary logistic regression, and controlling for potential confounding effects. The analysis for statistical association only focused on the first (2008) and last survey (2018). The odds ratio (OR) provides estimates of the likelihood or risk for different categories of the selected independent variables, relative to the reference category (RC). Statistical significance was determined at $p < 0.05$ for the bivariate analysis while the adjusted odds ratio (OR) and 95% confidence intervals (CI) were used for multivariable analysis. To account for variations in selection probability, we applied weighting factors at different levels of analysis. All analysis excluded missing responses and was conducted using Stata version 15.1

## Results

Table 2 presents the percentage distribution of socio-demographic characteristics. In the 2008, 2013 and 2018 surveys, the proportion of never-married but sexually active group–the interest group for this study–was higher among males (17.3–33.1%) compared to females (15.2–20.4%); among age group 20–24 (21.3–30.3%) compared to age group 15–19 (11.3–18.0); among young people with tertiary education (37.2–54.4%) or secondary education (20.5–32.0%) compared to those with none/primary education (4.3–7.0%); among Christians (29.7–35.4%) compared to Muslims (5.7–9.1%) or those of traditional and other religions (14.5–19.8%); among the young people in the richest wealth quintile (23.8–52.2%) compared to the poorest group (5.3–10.5%); and among urban residents (19.8–28.8%) compared to rural residents (12.4–21.1%). The proportion was far higher in southern region (27.0–45.7%) compared to norther regions (2.0–10.3%), except the North-Central (19.1–22.0%) which was slightly higher than other norther regions. The proportions who were not sexually active followed similar pattern with the never-married sexually active group.

The proportion of the young people who were ever married was highest among females (42.7–46.9%), age group 20–24 (48.8–57.6%), and among young people with none/primary education (62.1–66.3%), non-Christian religion (32.8–55.9%), the poorest wealth quintile (52.7–62.1%) and rural residence (42.0–45.9%). Northern regions in this case had higher proportions of the ever married young people (42.4–61.6%), except North-Central (33.1–34.2%) which is lower than other norther regions.

### Prevalence of risky sexual behaviours and trends

The prevalence of risky behaviour across the three surveys was 63.3–76.7% for inconsistent condom use, 57.9–72.4% for non-use of modern contraceptives, 53.3%– 55.2% for early sex, and 7.8–9.6% for multiple sexual partner (Table 3). Over four-fifths of unmarried young people in Nigeria engaged in at least one risk behaviours in each survey year. For each of the survey years and sexes, inconsistent condom use had the highest prevalence among the four risky SRH behaviours, followed by non-use of modern contraceptives while multiple sexual partner had the lowest prevalence. For each of the survey years, females generally a higher prevalence of each of the risk factors compared to males except in the case of multiple sexual partners. Overall, the proportion engaging in at least one risky sexual behaviour was 87.2% of females compared to 85.9% of males in 2008, 82.5% of females versus 80.4% of males in 2013, and 93.7% for females and 71.1% for males in 2018.

**Trends.** As Table 3 further shows, the pattern of changes in the four focal SRH risky behaviours was consistent over the 10-year period, with the highest rates of each sexual risk behaviour occurring in 2018 and the lowest rates in 2013. Overall, about 3–12% decline was recorded for the four SRH behaviours between 2008 and 2013, while a range of 8–25% increase

**Table 2. Socio-demographics distribution of the datasets and the respondents.**

| Socio-demographic characteristics | Study Period | | | | | | | | | | | |
|---|---|---|---|---|---|---|---|---|---|---|---|---|
| | 2008 (N = 17,509) | | | | 2013 (N = 21,056) | | | | 2018 (N = 19,146) | | | |
| **Sex** | Ever married n = 6,080 | Not sexually active n = 7,239 | Never-married sexually active n = 4,190 | Sub-total | Ever married n = 7,317 | Not sexually active n = 9,576 | Never-married sexually active n = 4,163 | Sub-total | Ever married n = 6,767 | Not sexually active n = 9,389 | Never-married sexually active n = 2,990 | Sub-total |
| Male | 8.1 | 58.9 | 33.1 | 4889 | 7.5 | 66.3 | 26.2 | 6496 | 6.3 | 76.4 | 17.3 | 3876 |
| Female | 45.1 | 34.6 | 20.4 | 12620 | 46.9 | 36.2 | 16.9 | 14560 | 42.7 | 42.1 | 15.2 | 15270 |
| **Age groups** | | | | | | | | | | | | |
| 15–19 | 21.4 | 60.6 | 18.0 | 9006 | 20.7 | 65.2 | 14.1 | 11419 | 18.3 | 70.4 | 11.3 | 10846 |
| 20–24 | 48.8 | 20.9 | 30.3 | 8503 | 51.5 | 22.1 | 26.5 | 9637 | 57.6 | 21.1 | 21.3 | 8300 |
| **Education** | | | | | | | | | | | | |
| None/Primary | 66.3 | 26.6 | 7.0 | 6634 | 64.0 | 30.4 | 5.6 | 8098 | 62.1 | 33.6 | 4.3 | 7153 |
| Secondary | 15.7 | 52.3 | 32.0 | 9797 | 17.0 | 56.8 | 26.2 | 11701 | 20.1 | 59.4 | 20.5 | 10648 |
| Tertiary | 12.9 | 32.8 | 54.4 | 1078 | 11.6 | 37.4 | 51.0 | 1257 | 13.5 | 49.3 | 37.2 | 1345 |
| **Religion** | | | | | | | | | | | | |
| Christianity | 19.2 | 45.4 | 35.4 | 9803 | 18.5 | 46.8 | 34.6 | 9839 | 21.1 | 49.2 | 29.7 | 7878 |
| Islam | 54.4 | 36.5 | 9.1 | 7457 | 49.2 | 44.4 | 6.4 | 10967 | 45.4 | 48.9 | 5.7 | 11172 |
| Traditional and others | 55.9 | 24.4 | 19.8 | 183 | 40.6 | 39.4 | 20.1 | 150 | 32.8 | 52.8 | 14.5 | 96 |
| **Wealth quintile** | | | | | | | | | | | | |
| Poorest | 62.1 | 27.4 | 10.5 | 2921 | 59.1 | 36.7 | 4.1 | 3514 | 52.7 | 42.0 | 5.3 | 3425 |
| Poorer | 49.4 | 34.9 | 15.7 | 3109 | 51.3 | 36.8 | 11.9 | 4018 | 49.3 | 40.8 | 9.8 | 3956 |
| Middle | 32.6 | 43.4 | 24.0 | 3481 | 31.2 | 46.6 | 22.3 | 4435 | 36.5 | 47.6 | 15.9 | 3937 |
| Rich | 24.0 | 44.2 | 31.8 | 4143 | 24.4 | 49.9 | 25.7 | 4620 | 25.1 | 52.8 | 22.1 | 4152 |
| Richest | 15.6 | 52.2 | 32.2 | 3855 | 15.0 | 54.5 | 30.5 | 4469 | 14.4 | 61.8 | 23.8 | 3676 |
| **Residence** | | | | | | | | | | | | |
| Urban | 22.0 | 49.1 | 28.8 | 6368 | 20.0 | 55.4 | 24.6 | 8976 | 21.7 | 58.5 | 19.8 | 8380 |
| Rural | 42.0 | 36.9 | 21.1 | 11141 | 45.7 | 38.1 | 16.2 | 12080 | 45.9 | 41.7 | 12.4 | 10766 |
| **Region** | | | | | | | | | | | | |
| North Central | 33.9 | 44.1 | 22.0 | 2694 | 33.1 | 45.4 | 21.5 | 3189 | 34.2 | 46.7 | 19.1 | 2789 |
| North East | 56.1 | 33.7 | 10.3 | 2166 | 50.6 | 40.8 | 8.6 | 3199 | 42.4 | 49.3 | 8.3 | 3436 |
| North West | 61.6 | 35.9 | 2.5 | 3933 | 50.9 | 45.4 | 3.7 | 6441 | 50.2 | 47.8 | 2.0 | 6218 |
| South East | 15.3 | 53.1 | 31.6 | 2186 | 14.5 | 51.3 | 34.2 | 2387 | 17.3 | 55.7 | 27.0 | 1955 |
| South South | 18.5 | 35.8 | 45.7 | 3150 | 16.2 | 40.9 | 42.9 | 2794 | 18.6 | 43.5 | 37.9 | 2005 |
| South West | 18.1 | 48.0 | 33.9 | 3380 | 18.6 | 50.3 | 31.1 | 3046 | 19.0 | 53.2 | 27.8 | 2743 |
| **Comprehensive HIV knowledge** | | | | | | | | | | | | |
| No | 38.6 | 40.8 | 20.6 | 11965 | 37.7 | 45.2 | 17.1 | 13973 | 34.6 | 52.3 | 13.1 | 11041 |
| Yes | 26.5 | 42.4 | 31.1 | 5544 | 29.0 | 46.0 | 25.0 | 7083 | 36.4 | 44.6 | 19.0 | 8105 |
| **Media exposure** | | | | | | | | | | | | |
| No | 49.5 | 35.0 | 15.5 | 6439 | 56.1 | 38.0 | 5.9 | 5095 | 47.6 | 45.0 | 7.4 | 6688 |
| Yes | 26.1 | 45.1 | 28.8 | 11070 | 27.9 | 47.9 | 24.2 | 15961 | 28.8 | 51.2 | 20.0 | 12457 |

was recorded between 2013 and 2018. Comparing 2008 directly with the 2018, the level of inconsistent condom use increased by 7% (from 72% to 77%) over the 10-year period, non-use of modern contraceptive increased by 10% (from 66% to 72%), having multiple sexual partners increased by 16% (from approximately 8% to 10%) and early sexual engagements increased by

4% (from 55% to 57%). Whereas the five-year period between the first two surveys (2008–2013) recorded a decrease in the prevalence of each SRH behaviour, the period between the last two surveys (2013–2018) recorded an increase for each indicator. The changes in prevalence rate for 2013–2018 was generally of greater magnitude compared to 2008–2013 period. Overall, the proportion of young people who engaged in at least one risky sexual behaviour only increased marginally from 86.7% to 88.7% over the 10-year period of 2008 to 2018.

Compared to their male counterparts, young females generally had a higher prevalence of three of the four SRH risk behaviours–inconsistent condom use, non-use of contraceptives, and early sex–but a lower proportion engaged with multiple sexual partners. Males and females show different patterns in the trends of the four key behaviours of focus in this study. Whereas females show an overall increase over the ten-year period of 2008 to 2018 for each of the four SRH behaviours, males show an overall decrease in three of the four behaviours, with multiple sexual partners as the only exception. Overall, the proportion of the sample that engaged in at least one risky sexual behaviour decreased from 85.9% to 71.1% between 2008 and 2018 (17.5% decrease) for males, but increased from 87.2 to 93.7% (7.5% increase) for females.

## Factors associated with risky sexual behaviours

**Bivariate analysis.** As shown in Table 4, with the exception of religion in both 2008 and 2018 and region of residence in 2018, all the key sociodemographic factors–age, level of education, wealth quintile, rural-urban residence, region of residence–were significantly associated with inconsistent condom use among the never-married young men in 2008 and 2018 (p<0.05). The associated socio-demographic factors among females were different in each survey year. While education and region of residence were significantly associated with inconsistent condom use among the young women in 2008 and 2018, other factors–age, wealth quintile, rural-urban residence–were significant only in 2008; except religion which showed no significant association in both survey years. Comprehensive HIV knowledge was also significantly associated with inconsistent condom use among males in 2008 and 2018 but was a significant factor among females only in 2008. Conversely, while media exposure showed a significant association among males only in 2018, it was significant among females in both 2008 and 2018.

The factors significantly associated with non-use of any modern contraceptive were similar to those of inconsistent condom use among the never-married young men in both 2008 and 2018, except religion which was significant in 2008 and media exposure which was significant only in 2018. The results of both non-use of any modern contraceptive and inconsistent condom use were also similar for the females, except religion which was significant in 2018. For multiple sexual partners, only two factors–age, and region–had a significant association among males and the pattern was the same for 2008 and 2018. Also, among the young women, only two factors–education in 2008 and region of residence in both 2008 and 2018 –were significantly associated with multiple sexual partners. All the explanatory variables were significantly associated with sexual debut before age 18 among females in both 2008 and 2018 except religion which was significant only in 2018. For males, while age, education and media exposure were significantly associated with sexual debut before age 18 in both 2008 and 2018 among the young men, all the other factors–wealth quintile, rural-urban residence, region of residence and media exposure–were significant only in 2008, except religion which had no significant association. Further, except rural-urban residence and region which were significant in 2008, media exposure in 2018 and religion which was not significant, all other factors–age, education, wealth quintile and comprehensive HIV knowledge were significantly associated

**Table 3. Changes in contraceptive and sexual behaviour of unmarried young people in Nigeria.**

| | Inconsistent condom use | Non-use of any modern contraceptive | Multiple sexual partner | Proportion having sexual debut before age 18 | At least one sexual risk |
|---|---|---|---|---|---|
| **Overall** | | | | | |
| 2008 (N = 4,216) | 71.5 | 66.0 | 8.3 | 55.2 | 86.7 |
| 2013 (N = 4,194) | 63.3 | 57.9 | 7.8 | 53.3 | 81.6 |
| 2018 (N = 3,015) | 76.7 | 72.4 | 9.6 | 57.4 | 88.6 |
| 5-year percentage change (2008–2013) | 11.5 ↓ | 12.3 ↓ | 6.0 ↓ | 3.4 ↓ | 12.3 ↓ |
| 5-year percentage change (2013–2018) | 21.2 ↑ | 25.0 ↑ | 23.1 ↑ | 7.7 ↑ | 8.6 ↑ |
| 10-year percentage change (2008–2018) | 7.3 ↑ | 9.7 ↑ | 15.7 ↑ | 4.0 ↑ | 2.2 ↑ |
| **Males** | | | | | |
| 2008 (N = 1,638) | 67.1 | 59.2 | 14.6 | 55.7 | 85.9 |
| 2013 (N = 1,719) | 61.7 | 53.2 | 13.4 | 48.9 | 80.4 |
| 2018 (N = 681) | 46.9 | 38.7 | 18.5 | 48.5 | 71.1 |
| 5-year percentage change (2008–2013) | 8.0 ↓ | 10.1 ↓ | 8.2 ↓ | 12.2 ↓ | 6.4 ↓ |
| 5-year percentage change (2013–2018) | 24.0 ↓ | 27.3 ↓ | 38.1 ↑ | 0.8 ↓ | 11.6 ↓ |
| 10-year percentage change (2008–2018) | 30.1 ↓ | 34.6 ↓ | 26.7 ↑ | 12.9 ↓ | 17.2 ↓ |
| **Females** | | | | | |
| 2008 (N = 2,578) | 74.3 | 70.3 | 4.3 | 54.9 | 87.2 |
| 2013 (N = 2,475) | 64.4 | 61.2 | 4.0 | 56.3 | 82.5 |
| 2018 (N = 2,334) | 85.5 | 82.7 | 7.0 | 60.0 | 93.7 |
| 5-year percentage change (2008–2013) | 13.3 ↓ | 12.9 ↓ | 7.0 ↓ | 2.6 ↑ | 5.4 ↓ |
| 5-year percentage change (2013–2018) | 32.8 ↑ | 35.1 ↑ | 75.0 ↑ | 6.6 ↑ | 13.6 ↑ |
| 10-year percentage change (2008–2018) | 15.1 ↑ | 17.6 ↑ | 62.8 ↑ | 9.3 ↑ | 7.5 ↑ |

**Note:** results are in percentages; ↑- Increase; ↓ decline.

with involvement in at least one sexual risk behaviour among males in both 2008 and 2018. The result was similar among female with age, education, wealth quintile, rural-urban residence, religion, region of residence and comprehensive HIV knowledge, except media exposure which was significant for both 2008 and 2018.

**Multivariable analysis.** Table 5 presents the results of the binary logistic regression. Among female gender, poorest wealth quintile (OR = 3.55; 95% C.I. = 1.79–7.05) and rural residence (OR = 1.61; 95% C.I. = 1.21–2.16) were significantly associated with increased odds of inconsistent condom use in 2008. Compared to females from the South-West, females from all other regions, except South-East and North-West, had significantly higher odds of inconsistent condom use: North-Central (OR = 1.53; 95% C.I. = 1.04–2.25), South-South (OR = 2.09; 95% C.I. = 1.47–2.98) and North-East (OR = 2.22; 95% C.I. = 1.24–3.97). Contrariwise, older age group (20–24 years) (OR = 0.64; 95% C.I. = 0.55–0.76), higher education (OR = 0.43; 95% C.I. = 0.27–0.71 for secondary education, and OR = 0.33; 95% C.I. = 0.19–0.56 for tertiary education) and comprehensive HIV/AIDS knowledge (OR = 0.72; 95% C.I. = 0.58–0.90) were significantly associated with 28–67% lower odds of inconsistent condom use in 2008 among

**Table 4. Factors associated with contraceptive and sexual behaviour of young people in Nigeria using 2008 and 2018 Survey.**

| Variables | Males | | | | | | | | | |
|---|---|---|---|---|---|---|---|---|---|---|
| | Inconsistent condom use | | Non-use of any modern contraceptive | | Multiple sexual partner | | Sexual debut before age 18 | | At least one sexual risk | |
| | 2008 | 2018 | 2008 | 2018 | 2008 | 2018 | 2008 | 2018 | 2008 | 2018 |
| **Age groups** | n (%) | n (%) | n (%) | n (%) | n (%) | n (%) | n (%) | n (%) | n (%) | n (%) |
| 15–19 | 415(77.2) | 122(53.1) | 389(72.1) | 104(45.2) | 59(11.0) | 28(12.3) | 459(85.4) | 192(83.8) | 515(95.9) | 203(88.5) |
| 20–24 | 684(62.1) | 197(43.7) | 583(52.9) | 160(35.4) | 179(16.3) | 98(21.6) | 454(41.2) | 138(30.6) | 891(81.0) | 281(62.2) |
| $\chi^2$ | 30.28*** | 4.15* | 47.44*** | 4.99* | 6.09* | 7.93** | 236.6*** | 131.90*** | 42.72*** | 36.92*** |
| **Education** | | | | | | | | | | |
| None/Primary | 180(88.3) | 48(68.) | 169(82.6) | 45(65.0) | 24(11.6) | 8(11.9) | 140(68.4) | 28(39.9) | 201(98.6) | 53(75.8) |
| Secondary | 821(67.0) | 226(45.7) | 726(59.3) | 185(37.4) | 175(14.3) | 94(19.0) | 705(57.6) | 276(55.7) | 1056(86.3) | 366(73.9) |
| Tertiary | 98(46.7) | 45(38.9) | 75(35.9) | 33(28.7) | 40(19.0) | 23(20.0) | 67(32.2) | 27(23.1) | 149(71.1) | 65(55.9) |
| $\chi^2$ | 29.75*** | 6.06** | 35.59*** | 10.52*** | 1.73 | 0.91 | 28.80*** | 16.36*** | 24.75*** | 6.00** |
| **Wealth quintile** | | | | | | | | | | |
| Poorest | 128(91.3) | 34(78.2) | 119(84.8) | 33(76.3) | 17(11.8) | 6(14.5) | 92.0(65.6) | 19(44.5) | 138(98.2) | 36(83.1) |
| Second | 157(76.4) | 46(56.2) | 141(68.7) | 41(49.8) | 27(12.9) | 20(23.7) | 127(62.8) | 44(53.4) | 187(91.2) | 68(82.8) |
| Middle | 238(72.3) | 82(55.8) | 216(65.5) | 70(47.2) | 48(14.5) | 22(14.7) | 194(58.8) | 74(49.9) | 287(87.0) | 108(73.3) |
| Fourth | 317(62.9) | 88(41.8) | 278(55.1) | 68(32.2) | 61(12.2) | 40(19.0) | 286(56.8) | 112(53.2) | 426(84.5) | 144(68.2) |
| Richest | 258(56.4) | 68(34.8) | 216(47.2) | 52(26.4) | 86(18.8) | 38(19.4) | 213(46.5) | 81(41.2) | 368(80.4) | 127(64.8) |
| $\chi^2$ | 15.74*** | 8.34*** | 16.51*** | 11.19*** | 2.22 | 0.76 | 5.34** | 1.27 | 7.33*** | 2.47* |
| **Residence** | | | | | | | | | | |
| Urban | 383(55.7) | 149(40.9) | 320(47.5) | 120(33.0) | 103(15.3) | 73(20.1) | 320(47.4) | 166(45.7) | 539(79.9) | 247(67.9) |
| Rural | 716(74.3) | 170(53.8) | 650(67.5) | 144(45.4) | 135(14.1) | 52(16.6) | 592(61.5) | 164(51.8) | 867(90.1) | 236(74.6) |
| $\chi^2$ | 35.94*** | 8.29** | 43.28*** | 7.73** | 0.32 | 0.98 | 23.71*** | 1.57 | 20.88*** | 2.21 |
| **Religion** | | | | | | | | | | |
| Christianity | 823(65.7) | 221(46.7) | 713(57.0) | 180(38.2) | 193(15.4) | 97(20.5) | 695(55.6) | 243(51.6) | 1066(85.1) | 343(72.5) |
| Islam and others | 267(71.2) | 99(47.2) | 249(66.5) | 84(40.0) | 43(11.5) | 29(13.9) | 211(56.3) | 87(41.6) | 329(87.9) | 141(67.7) |
| $\chi^2$ | 3.09 | 0.01 | 8.05** | 0.11 | 3.15 | 3.13 | 0.05 | 3.19 | 1.16 | 0.64 |
| **Region** | | | | | | | | | | |
| North Central | 213(73.3) | 50(48.9) | 196(67.3) | 43(42.4) | 41(14.1) | 12(11.8) | 174(59.6) | 42(41.4) | 265(91.0) | 71(69.6) |
| North East | 105(88.3) | 50(59.3) | 98(82.4) | 49(58.2) | 13(11.3) | 13(16.2) | 76(63.9) | 43(51.7) | 115(97.1) | 65(77.7) |
| North West | 44(79.6) | 21(51.1) | 42(75.3) | 15(36.1) | 3(4.6) | 6(14.2) | 23(41.4) | 19(44.8) | 48(86.5) | 31(73.1) |
| South East | 149(60.9) | 51(43.2) | 133(54.5) | 41(35.2) | 17(7.1) | 13(10.9) | 115(46.9) | 54(45.8) | 185(76.6) | 73(62.3) |
| South South | 351(74.3) | 84(45.4) | 295(62.4) | 66(35.5) | 94(19.9) | 62(33.5) | 284(60.1) | 107(58.1) | 432(91.4) | 146(78.9) |
| South West | 237(51.9) | 64(42.1) | 207(45.4) | 50(32.9) | 70(15.4) | 20(13.0) | 241(52.9) | 65(43.0) | 362(79.3) | 98(65.0) |
| $\chi^2$ | 14.18*** | 1.20 | 11.88*** | 2.50* | 4.48*** | 6.07*** | 3.74** | 1.59 | 10.98*** | 1.83 |
| **Compressive HIV knowledge** | | | | | | | | | | |
| Not comprehensive | 655(71.2) | 180(58.1) | 588(63.9) | 155(50.2) | 141(15.3) | 52(16.9) | 564(61.3) | 174(56.1) | 827(89.9) | 247(79.9) |
| Comprehensive | 443(61.8) | 139(37.5) | 382(53.2) | 108(29.2) | 98(13.6) | 74(19.8) | 348(48.5) | 157(42.2) | 580(81.7) | 237(63.7) |
| $\chi^2$ | 12.40** | 22.32*** | 14.59*** | 24.21*** | 0.65 | 0.86 | 21.81*** | 8.71** | 20.07*** | 13.79*** |
| **Media Exposure** | | | | | | | | | | |
| No | 239(61.8) | 68(63.8) | 221(57.2) | 55(52.1) | 63(16.4) | 24(22.3) | 213(55.1) | 52(49.2) | 326(84.1) | 87(82.4) |
| Yes | 860(68.7) | 252(43.7) | 749(59.9) | 208(36.3) | 175(14.0) | 102(17.8) | 699(55.9) | 278(48.4) | 1081(86.4) | 396(69.0) |
| $\chi^2$ | 4.37* | 12.72*** | 0.67 | 8.34** | 0.87 | 1.17 | 0.06* | 0.02 | 0.77 | 8.31** |
| | Females | | | | | | | | | |
| Variables | Inconsistent condom use | | Non-use of any modern contraceptive | | Multiple sexual partner | | Sexual debut before age 18 | | At least one sexual risk | |
| | 2008 | 2018 | 2008 | 2018 | 2008 | 2018 | 2008 | 2018 | 2008 | 2018 |
| **Age groups** | n (%) | n (%) | n (%) | n (%) | n (%) | n (%) | n (%) | n (%) | n (%) | n (%) |

(*Continued*)

**Table 4.** (Continued)

| | | | | | | | | | | |
|---|---|---|---|---|---|---|---|---|---|---|
| 15–19 | 878(79.9) | 884(87.2) | 845(77.1) | 859(84.7) | 51(4.7) | 58(5.7) | 948(86.3) | 877(86.5) | 1061(96.6) | 992(97.8) |
| 20–24 | 1038(70.2) | 1111(84.1) | 966(65.3) | 1071(81.1) | 58(4.0) | 104(7.9) | 467(31.6) | 534(39.7) | 1186(80.2) | 1196(90.6) |
| $\chi^2$ | 25.18*** | 2.79 | 35.67*** | 3.69* | 0.64 | 3.11 | 561.5*** | 423.82*** | 129.35*** | 38.94*** |
| **Education** | | | | | | | | | | |
| None/Primary | 240(90.1) | 221(91.3) | 236(88.6) | 217(89.6) | 15(5.8) | 20(8.3) | 183(68.9) | 197(81.4) | 256(96.2) | 239(98.6) |
| Secondary | 1446(74.7) | 1469(86.2) | 1377(71.2) | 1422(83.4) | 66(3.4) | 116(6.8) | 1149(59.4) | 1081(63.4) | 1721(88.9) | 1614(94.7) |
| Tertiary | 230(61.1) | 305(78.7) | 200(53.2) | 290(75.0) | 29(7.6) | 27(6.8) | 83(21.9) | 123(31.2) | 270(71.7 | 335(86.5) |
| $\chi^2$ | 26.44*** | 8.08** | 837.66*** | 8.78*** | 5.98** | 0.27 | 77.44*** | 73.93*** | 38.02*** | 20.93*** |
| **Wealth quintile** | | | | | | | | | | |
| Poorest | 160(94.4) | 129(90.8) | 157(92.5) | 127(88.8) | 13(8.0) | 10(7.0) | 123(72.4) | 102(71.8) | 168(98.8) | 138(96.9) |
| Second | 225(79.6) | 275(89.1) | 218(77.2) | 268(86.7) | 7(2.6) | 30(9.6) | 213(75.3) | 223(72.2) | 272(96.1) | 300(96.9) |
| Middle | 396(77.8) | 420(87.1) | 378(74.3) | 401(83.1) | 19(3.7) | 33(6.8) | 318(62.4) | 333(69.0) | 463(90.1) | 466(96.6) |
| Fourth | 613(74.5) | 597(83.3) | 572(69.6) | 576(80.5) | 38(4.6) | 58(8.1) | 424(51.5) | 428(59.8) | 714(86.8) | 652(91.1) |
| Richest | 521(65.7) | 573(83.8) | 487(61.4) | 558(81.7) | 32(4.0) | 32(4.6) | 338(42.6) | 315(46.1) | 630(79.5) | 631(92.4) |
| $\chi^2$ | 15.05*** | 1.97 | 16.18*** | 1.62 | 1.90 | 2.02 | 25.50*** | 13.68*** | 18.72*** | 5.58*** |
| **Residence** | | | | | | | | | | |
| Urban | 777(66.5) | 1124(85.7) | 729(62.4) | 1092(83.2) | 49(4.2) | 82(6.2) | 529(45.3) | 701(53.4) | 939(80.4) | 1218(92.8) |
| Rural | 1138(80.8) | 870(85.2) | 1084(76.9) | 837(81.9) | 61(4.3) | 81(7.9) | 886(62.8) | 701(68.6) | 1307(92.3) | 970(95.0) |
| $\chi^2$ | 33.70*** | 0.06 | 32.11*** | 0.37 | 0.03 | 1.75 | 42.20*** | 35.93*** | 58.12*** | 3.27 |
| **Religion** | | | | | | | | | | |
| Christianity | 1671(74.6) | 1596(84.5) | 1582(71.6) | 1536(81.3) | 100(4.5) | 143(7.6) | 1244(55.5) | 1105(58.5) | 1954(87.2) | 1761(93.2) |
| Islam and others | 240(74.0) | 389(89.5) | 228(70.4) | 384(88.3) | 9(2.9) | 20.0(4.5) | 160(49.4) | 288(66.2) | 281(86.6) | 418(96.1) |
| $\chi^2$ | 0.03 | 3.52 | 0.01 | 5.78* | 1.57 | 3.00 | 3.05 | 5.64* | 0.05 | 3.51 |
| **Region** | | | | | | | | | | |
| North Central | 239(78.4) | 382(88.7) | 233(76.3) | 374(86.7) | 10(3.4) | 21(4.9) | 168(55.0) | 289(67.0) | 275(90.2) | 413(95.8) |
| North East | 92(88.4) | 162(79.4) | 91(87.3) | 158(77.3) | 7(6.4) | 38(18.8) | 71(68.1) | 147(72.0) | 100(95.9) | 194(94.9) |
| North West | 32(71.9) | 67(80.1) | 32(71.9) | 66(78.9 | 2(6.4) | 2(2.7) | 19(43.8) | 52(61.4) | 39(88.0) | 75(89.7) |
| South East | 368(80.5) | 365(87.2) | 357(78.1) | 344(82.1) | 13(2.9) | 23(5.4) | 227(49.6) | 214(51.1) | 406(88.8) | 386(92.1) |
| South South | 726(74.4) | 465(80.1) | 662(67.9) | 447(76.9) | 60(6.1) | 54(9.3) | 618(63.4) | 372(64.0) | 879(90.1) | 536(92.2) |
| South West | 450(66.3) | 552(89.8) | 439(63.3) | 541(88.0) | 17(2.5) | 24(3.9) | 312(45.1) | 328(53.3) | 549(79.2) | 584(95.1) |
| $\chi^2$ | 6.23*** | 4.30*** | 7.04*** | 4.22** | 3.28** | 9.83*** | 10.11*** | 6.87*** | 9.66*** | 1.60 |
| **Compressive HIV knowledge** | | | | | | | | | | |
| Not comprehensive | 1207(77.4) | 1001(87.0) | 1150(73.8) | 965(83.8) | 65(4.2) | 79(6.9) | 893(57.3) | 760(66.0) | 1389(89.1) | 1085(94.3) |
| Comprehensive | 709(69.6) | 993(84.0) | 663(65.1) | 965(81.5) | 45(4.4) | 83(7.0) | 522(51.2) | 641(54.2) | 858(84.2) | 1103(93.2) |
| $\chi^2$ | 15.29*** | 3.41 | 18.19*** | 1.77 | 0.06 | 0.01 | 6.96* | 25.32*** | 7.89** | 0.99 |
| **Media Exposure** | | | | | | | | | | |
| No | 484(78.0) | 353(89.6) | 466(75.0) | 346(87.8) | 30(4.9) | 31(7.8) | 383(61.7) | 299(76.0) | 559(90.0) | 385(97.8) |
| Yes | 1431(73.2) | 1642(85.5) | 1348(68.9) | 1584(81.6) | 79(4.1) | 132(6.8) | 1032(52.7) | 1102(56.8) | 1688(86.3) | 1803(92.9) |
| $\chi^2$ | 4.16* | 4.70* | 6.10* | 6.58* | 0.68 | 0.37 | 10.44** | 38.64*** | 4.25* | 12.71*** |

\*\*\* p<0.001

\*\* p<0.01

\* p<0.05.

females. The factors associated with inconsistent condom use in 2018 were similar to those of 2008 except that age, secondary education, wealth quintile, rural-urban residence and media exposure were not significant, and most of the other regions had lower odds compared to the

**Table 5. Binary logistic regression showing factors associated with sexual behaviour among young people.**

| | Males | | | | | | | | | |
|---|---|---|---|---|---|---|---|---|---|---|
| **Variables** | Inconsistent condom use | | Any modern Contraceptive | | Multiple sexual partner | | Sexual debut before age 18 | | At least one sexual risk | |
| | Adjusted OR (95% C.I.) | | Adjusted OR (95% C.I.) | | Adjusted OR (95% C.I.) | | Adjusted OR (95% C.I.) | | Adjusted OR (95% C.I.) | |
| **Age groups** | 2008 | 2018 | 2008 | 2018 | 2008 | 2018 | 2008 | 2018 | 2008 | 2018 |
| 15–19 | 1.00 | 1.00 | 1.00 | 1.00 | 1.00 | 1.00 | 1.00 | 1.00 | 1.00 | 1.00 |
| 20–24 | 0.58*** (0.44–0.78) | 0.71 (0.48–1.05) | 0.53*** (0.41–0.68) | 0.68* (0.46–1.00) | 1.64* (1.12–2.39) | 2.23** (1.31–3.78) | 0.13*** (0.10–0.18) | 0.09*** (0.05–0.14) | 0.22*** (0.12–0.39) | 0.23*** (0.13–0.40) |
| **Education** | | | | | | | | | | |
| None/Primary | 1.00 | 1.00 | 1.00 | 1.00 | 1.00 | 1.00 | 1.00 | 1.00 | 1.00 | 1.00 |
| Secondary | 0.37*** (0.22–0.62) | 0.52* (0.27–0.99) | 0.44*** (0.29–0.68) | 0.41** (0.21–0.78) | 1.09 (0.64–1.86) | 1.54 (0.69–3.41) | 0.67* (0.46–0.99) | 1.20 (0.64–2.26) | 0.12*** (0.03–0.40) | 0.82 (0.40–1.67) |
| Tertiary | 0.26*** (0.14–0.48) | 0.62 (0.28–1.38) | 0.27*** (0.15–0.49) | 0.43* (0.19–0.95) | 1.32 (0.65–2.66) | 1.46 (0.57–3.79) | 0.47** (0.28–0.81) | 0.58 (0.25–1.35) | 0.08*** (0.02–0.32) | 0.64 (0.28–1.49) |
| **Wealth status** | | | | | | | | | | |
| Poorest | 2.40** (1.27–4.52) | 2.17 (0.89–5.28) | 1.67 (0.96–2.92) | 2.55* (1.10–5.89) | 0.85 (0.43–1.69) | 1.51 (0.57–3.99) | 0.92 (0.55–1.55) | 0.92 (0.40–211) | 3.70* (1.22–11.23) | 1.77 (0.64–4.89) |
| Poorer | 1.04 (0.67–1.63) | 0.85 (0.48–1.48) | 0.95 (0.63–1.43) | 0.94 (0.54–1.62) | 0.82 (0.47–1.41) | 2.10 (0.96–4.59) | 1.04 (0.68–1.60) | 1.10 (0.58–2.08) | 1.18 (0.62–2.26) | 1.57 (0.74–3.32) |
| Middle | 1.00 | 1.00 | 1.00 | 1.00 | 1.00 | 1.00 | 1.00 | 1.00 | 1.00 | 1.00 |
| Richer | 0.78 (0.55–1.10) | 0.66 (0.41–1.05) | 0.81 (0.57–1.13) | 0.60* (0.38–0.96) | 0.78 (0.49–1.26) | 1.15 (0.61–2.18) | 1.19 (0.84–1.68) | 1.39 (0.81–2.41) | 1.13 (0.67–1.91) | 0.87 (0.52–1.47) |
| Richest | 0.89 (0.60–1.32) | 0.55* (0.34–0.88) | 0.86 (0.59–1.26) | 0.56* (0.33–0.94) | 1.34 (0.81–2.21) | 0.89 (0.44–1.78) | 0.91 (0.61–1.34) | 1.11 (0.56–2.19) | 1.27 (0.73–2.22) | 0.91 (0.48–1.70) |
| **Region** | | | | | | | | | | |
| South West | 1.00 | 1.00 | 1.00 | 1.00 | 1.00 | 1.00 | 1.00 | 1.00 | 1.00 | 1.00 |
| North Central | 1.93** (1.30–2.86) | 1.00 (0.53–1.87) | 1.91*** (1.33–2.74) | 1.23 (0.63–2.40) | 1.12 (0.69–1.81) | 0.99 (0.44–2.23) | 1.02 (0.70–1.51) | 0.99 (0.46–2.15) | 2.07** (1.21–3.56) | 1.15 (0.52–2.57) |
| North East | 3.89*** (1.99–7.51) | 0.96 (0.48–1.93) | 3.24*** (1.83–5.74) | 1.51 (0.74–3.05) | 0.97 (0.53–1.75) | 1.35 (0.50–3.65) | 1.13 (0.69–1.85) | 1.70 (0.68–4.24) | 4.84*** (1.89–12.4) | 1.27 (0.50–3.18) |
| North West | 2.81** (1.23–6.41) | 1.15 (0.51–2.59) | 2.89** (1.42–5.86) | 0.91 (0.34–2.40) | 0.34 (0.07–1.56) | 1.08 (0.30–3.91) | 0.50 (0.27–0.95) | 1.58 (0.66–3.78) | 1.46 (0.55–3.88) | 1.40 (0.46–4.30) |
| South South | 1.50 (0.98–2.30) | 0.99 (0.57–1.73) | 1.56* (1.04–2.36) | 1.08 (0.60–1.94) | 0.42 (0.21–0.84) | 0.85 (0.34–2.015) | 0.72 (0.47–1.09) | 1.04 (0.49–2.22) | 0.78 (0.46–1.33) | 0.84 (0.43–1.65) |
| South East | 2.55*** (1.74–3.72) | 1.07 (0.60–1.91) | 1.87*** (1.30–2.70) | 1.15 (0.62–2.14) | 1.48 (0.91–2.42) | 3.90** (1.71–8.87) | 1.13 (0.78–1.64) | 2.35* (1.13–4.85) | 2.55*** (1.52–4.26) | 2.09* (1.05–4.12) |
| **Residence** | | | | | | | | | | |
| Urban | 1.00 | 1.00 | 1.00 | 1.00 | 1.00 | 1.00 | 1.00 | 1.00 | 1.00 | 1.00 |
| Rural | 1.25 (0.91–1.72) | 1.14 (0.76–1.71) | 1.44* (1.06–1.96) | 1.04 (0.68–1.59) | 1.05 (0.68–1.61) | 0.47** (0.27–0.83) | 1.27 (0.95–1.71) | 1.07 (0.66–1.73) | 0.57** (0.39–0.83) | 0.87 (0.57–1.35) |
| **Religion** | | | | | | | | | | |
| Christianity | 1.00 | 1.00 | 1.00 | 1.00 | 1.00 | 1.00 | 1.00 | 1.00 | 1.00 | 1.00 |
| Islam and others | 1.30 (0.91–1.85) | 0.76 (0.47–1.24) | 1.44* (1.01–2.04) | 0.73 (0.44–1.21) | 0.75 (0.48–1.16) | 0.81 (0.42–1.59) | 0.99 (0.73–1.37) | 0.83 (0.46–1.51) | 1.07 (0.65–1.79) | 0.75 (0.40–1.41) |
| **Comprehensive HIV/ AIDS knowledge** | | | | | | | | | | |
| Not comprehensive | 1.00 | 1.00 | 1.00 | 1.00 | 1.00 | 1.00 | 1.00 | 1.00 | 1.00 | 1.00 |
| Comprehensive | 0.76* (0.58–0.98) | 0.52** (0.36–0.76) | 0.77* (0.60–1.00) | 0.51** (0.35–0.75) | 0.78 (0.57–1.09) | 1.13 (0.71–1.81) | 0.74* (0.58–0.95) | 0.54** (0.34–0.84) | 0.57** (0.39–0.83) | 0.51** (0.31–0.82) |
| **Exposure to mass media** | | | | | | | | | | |
| No | 1.00 | 1.00 | 1.00 | 1.00 | 1.00 | 1.00 | 1.00 | 1.00 | 1.00 | 1.00 |
| Yes | 1.43* (1.05–1.95) | 0.68 (0.38–1.20) | 1.14 (0.85–1.52) | 0.96 (0.57–1.60) | 0.87 (0.58–1.09) | 0.66 (0.37–1.19) | 1.17 (0.86–1.59) | 1.80* (1.05–3.09) | 1.33 (0.88–1.99) | 0.83 (0.46–1.50) |

(*Continued*)

**Table 5.** (*Continued*)

| | Females | | | | | | | | | |
|---|---|---|---|---|---|---|---|---|---|---|
| **Variables** | Inconsistent condom use | | Any modern Contraceptive | | Multiple sexual partner | | Sexual debut before age 18 | | At least one sexual risk | |
| | Adjusted OR (95% C.I.) | | Adjusted OR (95% C.I.) | | Adjusted OR (95% C.I.) | | Adjusted OR (95% C.I.) | | Adjusted OR (95% C.I.) | |
| **Age groups** | 2008 | 2018 | 2008 | 2018 | 2008 | 2018 | 2008 | 2018 | 2008 | 2018 |
| 15–19 | 1.00 | 1.00 | 1.00 | 1.00 | 1.00 | 1.00 | 1.00 | 1.00 | 1.00 | 1.00 |
| 20–24 | 0.74** (0.60–0.91) | 0.89 (0.66–1.20) | 0.71*** (0.58–0.87) | 0.87 (0.67–1.14) | 0.67 (0.43–1.06) | 1.53* (1.04–2.26) | 0.09*** (0.07–0.11) | 0.12*** (0.09–0.15) | 0.19*** (0.12–0.28) | 0.27*** (0.15–0.46) |
| **Education** | | | | | | | | | | |
| None/Primary | 1.00 | 1.00 | 1.00 | 1.00 | 1.00 | 1.00 | 1.00 | 1.00 | 1.00 | 1.00 |
| Secondary | 0.43*** (0.27–0.71) | 0.71 (0.40–1.26) | 0.43*** (0.27–0.68) | 0.70 (0.40–1.22) | 0.69 (0.32–1.47) | 0.87 (0.47–1.59) | 1.07 (0.74–1.54) | 0.58* (0.38–0.08) | 0.57 (0.26–1.23) | 0.38 (0.13–1.14) |
| Tertiary | 0.33*** (0.19–0.56) | 0.49* (0.26–0.93) | 0.28*** (0.16–0.47) | 0.46* (0.25–0.84) | 2.28 (0.86–6.03) | 0.81 (0.39–1.69) | 0.50*** (0.31–0.82) | 0.29** (0.17–0.48) | 0.38* (0.17–0.86) | 0.20** (0.06–0.62) |
| **Wealth status** | | | | | | | | | | |
| Poorest | 3.55*** (1.79–7.05) | 1.48 (0.67–3.27) | 3.14*** (1.83–5.39) | 1.55 (0.72–3.37) | 1.88 (0.86–4.11) | 0.70 (0.33–1.46) | 1.25 (0.76–2.05) | 0.84 (0.45–1.55) | 5.75* (1.29–25.54) | 0.65 (0.19–2.24) |
| Poorer | 1.01 (0.70–1.45) | 1.24 (0.72–2.13) | 1.09 (0.75–1.58) | 1.30 (0.81–2.09) | 0.65 (0.28–1.50) | 1.21 (0.70–2.12) | 1.55* (1.02–2.35) | 1.00 (0.69–1.46) | 1.99 (0.98–4.02) | 0.90 (0.40–2.02) |
| Middle | 1.00 | 1.00 | 1.00 | 1.00 | 1.00 | 1.00 | 1.00 | 1.00 | 1.00 | 1.00 |
| Richer | 1.15 (0.83–1.59) | 0.72 (0.49–1.06) | 1.11 (0.81–1.51) | 0.82 (0.58–1.15) | 1.21 (0.61–2.43) | 1.25 (0.76–2.07) | 0.85 (0.61–1.18) | 0.96 (0.69–1.34) | 1.13 (0.73–1.73) | 0.42** (0.24–0.75) |
| Richest | 1.04 (0.73–1.47) | 0.80 (0.49–1.28) | 1.04 (0.74–1.47) | 0.95 (0.62–1.44) | 0.92 (0.45–1.87) | 0.70 (0.37–1.31) | 0.82 (0.56–1.18) | 0.67* (0.46–0.96) | 1.01 (0.64–1.58) | 0.66 (0.35–1.22) |
| **Region** | | | | | | | | | | |
| South West | 1.00 | 1.00 | 1.00 | 1.00 | 1.00 | 1.00 | 1.00 | 1.00 | 1.00 | 1.00 |
| North Central | 1.53* (1.04–2.25) | 0.72 (0.43–1.22) | 1.56* (1.08–2.26) | 0.77 (0.48–1.25) | 1.45 (0.67–3.14) | 1.31 (0.57–3.00) | 1.33 (0.92–1.94) | 1.09 (0.74–1.61) | 2.01*** (1.31–3.09) | 0.76 (0.38–1.54) |
| North East | 2.22** (1.24–3.97) | 0.29** (0.14–0.59) | 2.35** (1.35–4.08) | 0.33*** (0.17–0.65) | 2.18 (0.84–5.70) | 5.93*** (2.95–11.90) | 2.13** (1.28–3.54) | 1.40 (0.83–2.36) | 3.35* (1.14–9.9) | 0.48 (0.15–1.52) |
| North West | 1.20 (0.50–2.88) | 0.46* (0.26–0.84) | 1.37 (0.57–3.30) | 0.53* (0.29–0.97) | 2.71 (0.62–11.8) | 0.63 (0.15–2.76) | 0,64 (0.31–1.31) | 1.46 (0.65–3.30) | 1.62 (0.66–4.02) | 0,40 (0.16–1.03) |
| South South | 2.09*** (1.47–2.98) | 0.79 (0.45–1.33) | 2.04*** (1.43–2.91) | 0.64 (0.39–1.06) | 1.33 (0.52–3.39) | 1.18 (0.62–2.22) | 1.26 (0.88–1.80) | 0.88 (0.60–1.28) | 2.22*** (1.46–3.39) | 0.55 (0.27–1.11) |
| South East | 1.28 (0.93–1.76) | 0.47** (0.29–0.75) | 1.04 (0.77–1.40) | 0.49** (0.32–0.76) | 2.87*** (1.50–5.48) | 2.36** (1.35–4.11) | 1.98*** (1.42–2.75) | 1.47* (1.01–2.15) | 2.02*** (1.37–2.99) | 0.53* (0.29–0.97) |
| **Residence** | | | | | | | | | | |
| Urban | 1.00 | 1.00 | 1.00 | 1.00 | 1.00 | 1.00 | 1.00 | 1.00 | 1.00 | 1.00 |
| Rural | 1.61*** (1.21–2.16) | 0.93 (0.89) | 1.50** (1.15–1.96) | 0.88 (0.63–1.23) | 0.89 (0.54–1.46) | 0.85 (0.53–1.36) | 1.13 (0.87–1.47) | 1.32 (1.01–1.73) | 1.76*** (1.26–2.45) | 1.18 (0.75–1.85) |
| **Religion** | | | | | | | | | | |
| Christianity | 1.00 | 1.00 | 1.00 | 1.00 | 1.00 | 1.00 | 1.00 | 1.00 | 1.00 | 1.00 |
| Islam and others | 1.20 (0.81–1.80) | 1.24 (0.77–1.99) | 1.11 (0.78–1.59) | 1.31 (0.83–2.08) | 0.90 (0.43–1.88) | 0.68 (0.35–1.33) | 0.95 (0.66–1.37) | 1.52* (1.04–2.22) | 1.45 (0.87–1.70) | 1.43 (0.71–2.87) |
| **Comprehensive HIV/ AIDS knowledge** | | | | | | | | | | |
| Not comprehensive | 1.00 | 1.00 | 1.00 | 1.00 | 1.00 | 1.00 | 1.00 | 1.00 | 1.00 | 1.00 |
| Comprehensive | 0.72** (0.58–0.90) | 0.91 (0.68–1.22) | 0.72** (0.59–0.89) | 1.01 (0.78–1.32) | 0.97 (0.60–1.57) | 1.04 (0.70–1.55) | 1.01 (0.81–1.26) | 0.87 (0.69–1.10) | 0.74 (0.53–1.03) | 1.29 (0.88–1.88) |

(*Continued*)

**Table 5.** (Continued)

| Exposure to mass media | | | | | | | | | | |
|---|---|---|---|---|---|---|---|---|---|---|
| No | 1.00 | 1.00 | 1.00 | 1.00 | 1.00 | 1.00 | 1.00 | 1.00 | 1.00 | 1.00 |
| Yes | 0.91 (0.70–1.20) | 0.73 (0.45–1.16) | 0.91 (0.70–1.18) | 0.69 (0.45–1.05) | 0.83 (0.50–1.39) | 1.21 (0.79–1.87) | 0.77 (0.59–1.01) | 0.69* (0.49–0.96) | 0.87 (0.60–1.24) | 0.44 (0.19–1.04) |

*** p<0.001

** p<0.01

* p<0.05.

South-West: North-East (OR = 0.29; 95% C.I. = 0.14–0.59), South-East (OR = 0.47; 95% C.I. = 0.29–0.75) and North-West (OR = 0.46; 95% C.I. = 0.26–0.84).

The pattern of association between inconsistent condom use and age, education, poorest wealth status and comprehensive HIV knowledge among males in 2008 was similar to that of the females. However, unlike the result among females in 2008, the odds of inconsistent condom use was significantly higher among males in North-West (OR = 2.81; 95% C.I. = 1.23–6.41) and South-East regions (OR = 2.55; 95% C.I. = 1.74–3.72) compared to the South-West, and among males with media exposure (OR = 1.43; 95% C.I. = 1.05–1.95). All the factors associated with inconsistent condom use among the males in 2008 were insignificant in 2018 except secondary education, and richest wealth quintile.

For non-use of any modern contraceptive among females, the odds was higher in 2008 for those in the poorest wealth quintile (OR = 3.14; 95% C.I. = 1.83–5.39); among rural dwellers (OR = 1.50; 95% C.I. = 1.15–1.96); and in the North-Central (OR = 1.56; 95% C.I. = 1.08–2.26), North-East (OR = 2.35; 95% C.I. = 1.35–4.08) and South-South (OR = 2.04; 95% C.I. = 1.43–2.91) compared to those in their respective reference categories. Conversely, the odds of non-use of modern contraceptives were significantly lower for females in older age group, those with higher education and for those with comprehensive HIV knowledge. In 2018, all the associated factors were not significant except that tertiary education (OR = 0.46; 95% C.I. = 0.25–0.84), residence in the North-East (OR = 0.33; 95% C.I. = 0.17–0.65), North-West (OR = 0.53; 95% C.I. = 0.29–0.97) and South-East (OR = 0.49; 95% C.I. = 0.32–0.76) were significantly associated with about 47–67% lower odds of non-use of any modern contraceptive compared to their respective reference groups.

The patterns of the association between non-use of any modern contraceptive and age, education, rural-urban residence, region and comprehensive HIV knowledge among males in 2008 were similar to that of the females in the same year, except wealth status which was not significant and religion (higher odds for Islamic religion) and two additional regions (North-West and South-East) which were significantly associated with higher odds of non-use of modern contraceptives relative to their respective reference groups. In 2018, while age, education and comprehensive HIV knowledge maintained the associations, all other previously associated factors were not significant except that, in addition, the odds of non-use of modern contraceptives was higher for the poorest wealth status (OR = 2.55; 95% C.I. = 1.10–5.89) and lower for the richer (OR = 0.60; 95% C.I. = 0.38–0.96) and richest status (OR = 0.56; 95% C.I. = 0.33–0.94) compared to the middle group.

For multiple sexual partners, while age was the only significant factor among males, region was the only factor among females in 2008. Older age 20–24 (OR = 1.64; 95% C.I. = 1.12–2.39) was associated with higher odds of multiple sexual partners compared to younger age 15–19 among males in 2008. In the same year, the odds of multiple sexual partners among males was higher in South-East (OR = 2.87; 95% C.I. = 1.50–5.48) compared to the South-West. In 2018,

while the odds of multiple sexual partners was higher for males with older age 20–24 (OR = 2.33; 95% C.I. = 1.31–3.78) and in South-East (OR = 3.90; 95% C.I. = 1.71–8.87), it was lower for rural residence (OR = 0.47; 95% C.I. = 0.27–0.83) compared to the corresponding reference groups. The pattern was similar to that of males in 2018, except that rural-urban residence was not significant and North-East was significantly associated with higher odds of multiple sexual partners among the females.

In 2008 and 2018, early sexual debut before age 18 was significantly associated with age and comprehensive HIV knowledge among males. For age, older males (20–24 years) had 87–99% lower odds for early sexual debut compared to younger males (15–19 years) in both 2008 (OR = 0.13; 95% C.I. = 0.10–0.18) and 2018 (OR = 0.09; 95% C.I. = 0.05–0.14). For HIV knowledge, comprehensive knowledge was associated with lower odds in 2008 (OR = 0.74; 95% C.I. = 0.58–0.95) and 2018 (OR = 0.54; 95% C.I. = 0.34–0.84). Other factors significantly associated with early sexual debut among males in 2008 was education (higher education had lower odds compared to no/primary education), and wealth index (richest group having lower odds compared with the poorest group), while the other significant factors in 2018 were region (South-East had higher odds) and media exposure (those exposed to the media had higher odds) compared to their corresponding reference groups. The pattern of associations among females was similar to that of the males in 2008 and 2018 for age, education and South-East region. Other associated factors among females dissimilar to males were wealth index (the poorest had higher odds in 2008 and the richest had lower odds in 2018), region (North-East had higher odds in 208), religion (Islam and others had higher odds in 2018) and media exposure (those with media exposure had lower odds in 2018).

For males, being involved in at least one sexual risk behaviour was associated with age, region and comprehensive knowledge of HIV/AIDS in 2008 and 2018. Older males had lower odds of at least one of the sexual risks in 2008 (OR = 0.22; 95% C.I. = 0.12–0.39) and 2018 (OR = 0.23; 95% C.I. = 0.13–0.40), males in the South-East had higher odds in 2008 (OR = 2.55; 95% C.I. = 1.52–4.26) and 2018 (OR = 2.09; 95% C.I. = 1.05–4.12), and comprehensive HIV knowledge had lower odds in 2008 (OR = 0.57; 95% C.I. = 0.39–0.83) and 2018 (OR = 0.51; 95% C.I. = 0.31–0.82) relative to their corresponding reference groups. While the pattern of association among females was similar to that of males for age and South-East residence in 2008 and 2018, education and wealth quintile were also significant factors among females in the same years. In addition, residence in rural area, North-Central, North-East and South-South was significantly associated with at least one sexual risk behaviour among females in 2008.

## Discussion

Young people in low- and middle-income countries (LMICs) continue to face health and development challenges, and young people in Africa have disproportionately high proportion of global deaths [1]. However, research on adolescent health challenges has lagged in Africa and programme efforts remain inadequate [25, 26]. These scenarios hold true for Nigeria and current studies are needed to understand the state of youth and adolescent sexual health and to contribute to the monitoring of progress regarding SDG target 3.7 (which aims to ensure universal access to SRH care services by 2030). Understanding young people's SRH is important, considering that they constitute a large proportion of vulnerable population in Nigeria. Besides, young people's SRH status has implication for the overall national health and development and the prospect for demographic dividends. This study addresses the identified evidence gap by exploring changes in contraceptive and sexual behaviours of unmarried sexually active young men and women in Nigeria.

Despite several implemented programmes which aimed at improving SRH behaviours and outcomes in Nigeria [11, 27, 28], our results show that more than four-fifths of young people in Nigeria consistently engaged in one or more risky sexual behaviour during the 2008–2018 period. We found a high prevalence of early sexual initiation, multiple sexual partnerships, non-use of modern contraceptives and inconsistent condom use over the ten-year study period.

Broadly, our results showed a higher level of risky sexual behaviour (inconsistent condom use, non-use of modern contraceptives, multiple sexual partnership and early sexual initiation) among the young people in 2018 compared to 2008. This finding strongly suggests that young people aged 15–24 in Nigeria have continued to have limited access to quality family planning and reproductive health services as reported in earlier studies [22, 29]. Although Nigeria national health policies stipulate that contraceptive service be made free to all people at the public health facilities, issues such as unfriendly attitudes of health care providers and weak contraceptive logistic systems continue to pose a challenge to young people's access to contraceptive and other relevant SRH services [29, 30]. As previously established in other SSA countries [19, 31, 32], non-use of contraceptives and erratic condom use among young people are perhaps due to unmet need for contraception, lack of youth friendly services, peer-pressure and personal preferences. These factors have implication and potential for sustaining high population growth in Nigeria which has a high-fertility context. Moreover, scholars have adduced that unintended pregnancy in adolescence is a major cause of adverse social and health outcomes, including unsafe abortion, lower educational achievements, low employment opportunities and reduced labour income in adulthood [33].

Our findings show that prevalence of risky sexual behaviour was generally higher among young women compared to men. This reflects a gender-based dimension in the poor SRH outcomes among young people and perhaps increase in sexual and gender violence in the recent years. Amo-Adjei and Tuoyire [20], in their study of timing of sexual debut among unmarried young people in SSA, also found a higher prevalence of early sexual debut among young women compared to men. Additionally, our trend analysis showed a considerably different pattern between males and females in the four indicators of sexual and contraceptive behaviours studied in this research. Whereas females show an overall increase in the 2008–2018 period for each of the four SRH behaviours, males had an overall decrease in three of the four behaviours, with multiple sexual partners as the only exception. Essentially, studies have highlighted that young women are more likely to engage in risky sexual behaviours with older men than with their male contemporaries [34, 35] as in the case of sugar daddies and "*blessers*" phenomenon reported in a study by Hoss and Blokland [36]. This factor partly accounts for a higher prevalence of HIV/AIDS among young women than men of same age due to age-disparate sexual partnerships [35]. The only exception we found in terms of the pattern of higher risky sexual behaviour among young women compared to their male peers was in respect of multiple sexual partnerships. Across the different data waves analysed in our study, we found that young women had a lower likelihood of having multiple sexual partners compared to their male counterparts.

Our results also showed a difference in sexual behaviour between adolescents (15–19 years) and youths (20–24), reflecting changes as transition takes place from younger to older phase of young lives, and indicating the need to target the sub-groups differently in programmatic focus. We found that youths (20–24 years) were less likely to report inconsistent condom use or non-use of modern method of contraception than adolescents (15–19 years), whereas youths were more likely to have multiple sexual partners than adolescents, reflecting the effect of age dynamics on sexual behaviour. Further, across different data points, youths reported having lower odds of early sexual initiation than those aged 15–19. This, perhaps, reflects an intergenerational shift in social norms and cohort effect that relate to sexual debut. In the

traditional Nigerian society, premarital sex is viewed as a taboo, but this social norm is fast changing among young people in the contemporary time.

Our analysis shows that region of residence is an important factor in sexual behaviour of young people. First, it is important to note that less than one-third of our study sample were young people resident in northern regions of Nigeria. Our focus on unmarried young people accounts for this as the rate of early marriage is quite high in northern Nigeria as established in previous studies [37–40]. Our findings show that North-east region had a significantly higher rate of multiple sexual partnerships compared to South-west, and young people in the South-west region were less likely to report inconsistent condom use compared to those in other regions of the country. These findings reflect the effects of residential spaces and place-based characteristics on sexual behaviours of young people [22]. Also our results show that young people in rural areas were more likely to report inconsistent condom use than those in urban settings, thus demonstrating that rural population remains underserved in family planning and reproductive health services.

The different data points in our analysis established comprehensive HIV/AIDS knowledge as a protective factor against early sex, inconsistent condom use and non-use of modern contraception among young people. Given that young people largely underestimate their risk of contracting HIV infection [38, 41, 42], there is need for expansion of programme on comprehensive HIV/AIDS knowledge among the youths.

Although the use of three repeated nationally representative sample surveys provide some strengths and it enabled us to explore trends and changes in contraceptive and sexual behaviours, secondary analysis of cross-sectional data has some limitations. This include inability to explore other relevant variables (outside the data collection framework of the original study) such as cultural practices and gender inequality dynamics which may affect SRH concerns of the adolescents. Besides, the study analysed self-reported data which may have some likelihood of social desirability and recall biases, albeit we used the most current information which offers the advantage and potential of reducing the possibility of recall bias.

## Conclusion

This study has provided an important up-to-date evidence on the changes in contraceptive and sexual behaviours of young unmarried men and women in Nigeria. The study established a consistently high prevalence of risky sexual behaviour among young people over the ten-year period (2008–2018). Overall, we found that the rate of inconsistent condom use, non-use of modern contraceptives, multiple sexual partnership and early sexual initiation remain high among Nigerian youths. Females constitute higher proportion with inconsistent condom use, non-use of modern contraceptives and early sexual initiation, while males constitute higher proportion with multiple sexual partnerships and no significant progress has been made in terms of contraceptive and sexual behaviours.

The present study lends credence to the previous findings and confirms Nigeria as a high-burden country for adolescent and youth health problems with SRH being a significant contributor. We found that prevalence of risky sexual behaviour was higher among young women than men, except for multiple sexual partnership, thus reflecting a gender-based dimension in the poor SRH outcomes. Our findings have important policy implications. There is need for gender-specific policies and programmes that account for specific gender peculiarities regarding the SRH outcomes between males and females. This study suggests that many previous policies and intervention efforts targeting adolescent and youth SRH have been largely ineffective, and therefore recommends the need to prioritise future actions that are well-tailored towards the specific needs of young men and women.

Considering the lack of progress towards achieving SDG target 3.7 as it relates to young people, there is need to examine the framing, coverage and effectiveness of SRH programming for young people for an improved outcome in Nigeria. The findings of the present study have utility for informing future actions and for guiding key areas for programmatic focus among the sub-groups of young people–male and female, as well as older and younger sub-groups.

## Author Contributions

**Conceptualization:** Sunday A. Adedini.

**Formal analysis:** Jacob Wale Mobolaji, Matthew Alabi.

**Methodology:** Sunday A. Adedini, Jacob Wale Mobolaji.

**Supervision:** Sunday A. Adedini, Adesegun O. Fatusi.

**Writing – original draft:** Sunday A. Adedini, Jacob Wale Mobolaji, Matthew Alabi, Adesegun O. Fatusi.

**Writing – review & editing:** Sunday A. Adedini, Jacob Wale Mobolaji, Matthew Alabi, Adesegun O. Fatusi.

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
