## [Decision Letter · Decision Letter 0]

25 Sep 2020

PONE-D-20-27905

Changes in contraceptive and sexual behaviours among unmarried young people in Nigeria: evidence from nationally representative surveys

PLOS ONE

Dear Dr. Adedini,

Thank you for submitting your manuscript to PLOS ONE. After careful consideration, we feel that it has merit but does not fully meet PLOS ONE’s publication criteria as it currently stands. Therefore, we invite you to submit a revised version of the manuscript that addresses the points raised during the review process.

Both reviewers make interesting points regarding the analysis and interpretation of results, which technically is sound. Look, in particular, at comments by reviewer 2, and please adjuts the text accordingly.

Expanding on reviewer 2 comments, there is a change in the analysis that I see necessary. Reviewer 2 points to the difference between males and females, and looking at table 4 we see that behaviour of males and females is radically different. However, the analysis is carried out jointly for both sexes with sex only as a control variable. I don't see this as appropriate. You should fit a different equation for each sex, or, if you prefer so, use a model with full interactions among sex and the rest of variables. That would allow you to test equality of the different coefficients according to sex. It is very difficult to interpret the results from the joint analysis and I don't see a priori any reasons why the effects of covariates should be the same for both sexes when levels are so different. As you know, one of the requirement for publication in Plos One is a high technical standard and this requirement goes in that direction.

One other minor comment: Table 4 does not specify if N is weighted or unweighted. Also there is no information on the kind of weighting applied. It should be normalized weights (average 1).

We look forward to receiving your revised manuscript.

Kind regards,

José Antonio Ortega, Ph.D.

Academic Editor

PLOS ONE

Journal Requirements:

'This work was partially supported by the Consortium for Advanced Research Training in Africa (CARTA), Nairobi, Kenya. The funders had no role in study design, data collection and analysis, decision to publish, or preparation of the manuscript.'

a. Please provide an amended statement that declares *all* the funding or sources of support (whether external or internal to your organization) received during this study, as detailed online in our guide for authors at http://journals.plos.org/plosone/s/submit-now

Please also include the statement “There was no additional external funding received for this study.” in your updated Funding Statement.

3. Please include your tables as part of your main manuscript and remove the individual files.

Please note that supplementary tables should be uploaded as separate "supporting information" files.

Reviewers' comments:

Reviewer's Responses to Questions

**Comments to the Author**

1. Is the manuscript technically sound, and do the data support the conclusions?

Reviewer #1: Yes

Reviewer #2: Yes

2. Has the statistical analysis been performed appropriately and rigorously? 

Reviewer #1: Yes

Reviewer #2: Yes

3. Have the authors made all data underlying the findings in their manuscript fully available?

Reviewer #1: Yes

Reviewer #2: Yes

4. Is the manuscript presented in an intelligible fashion and written in standard English?

Reviewer #1: Yes

Reviewer #2: Yes

5. Review Comments to the Author

Reviewer #1: The authors have contributed to an important aspect of sexual and reproductive health (SRH) by looking at the prevailing factors contributing to poor SRH outcomes. The novelty of this work is in no doubt considering the conceptualization, methodology and analysis. However, the author would have employed multi-level analysis; this would have shown the degree of variation within each community of the young people; also, this would have shown other factors that contribute to poor SRH outcomes among young Nigeria as such developing required interventions to address the study conclusion will be targeted to the needed group(s) at different levels, as against socio-demographic characteristics of the respondents alone. Though the technicality of this work is sound, this could be a very good suggestion to employ in the future.

Please kindly check on the manuscript submitted, from lines 56 to 58 about Nigeria being the country with the highest-burden of HIV - You might want to check the literature to be sure.

Also, please kindly check on the manuscript from lines 77 to 81 statement. It appears vague and not passing any message.

Reviewer #2: The manuscript is interesting and has presented the main idea clearly. The subject of this document is relevant and is within the interests of several international organizations that aim to reduce the risks associated to sexual risk behavior in order to achieve SRH-focused objectives. I think the document is on the right track, but it needs major revisions especially on the policy implications.

I include first a couple of comments at the general level and then some more specific ones.

General comments

1. Authors focus their research on the interest on SRH, as it is shown in the Introduction. However, the Discussion has lack of analysis regarding the policy implications and the agenda for policymakers. I expand this comment below.

2. Early sex is defined as having had first sexual intercourse at or before age 14. This is not necessarily the case in this paper. Maybe the authors could make it explicit and show the proportion of the sample falling into this definition. Beyond that, to generalize, author should avoid the term “early sex” and use “sexual debut during adolescence” or “sexual debut before age 18” when referring to this SRH dimension.

3. Nigeria is one of the most populated countries in the world and also one of the highest TFR. Even more, Nigeria is still on the early stage of fertility transition. Besides the potential risk of STIs due to the erratic use of condoms, non-use of modern contraceptives contributes to high levels of fertility. There is research discussing on the benefits of population reduction. Fertility preferences and desires could shape contraceptive use behavior. Authors should account for the high-fertility context of Nigeria at the moment of drawing their conclusions.

4. Author use the terms unmarried and never married as synonyms, although there is a subtle difference. Unmarried includes 4 categories: never married, widowed, not living together, and divorced. It is true that among women aged 15-24 never married and unmarried are quite similar; however, authors should note this difference for further research.

5. Minor language revision and proof-reading before resubmitting.

Specific comments

Introduction

6. From SRH perspective, one of the biggest concerns is unmet need for contraceptives. On the one hand, causes of unmet need are highly correlated to SRH rights violations (barriers to use, gender violence, and social pressure, among others) (e.g., Machiyama et al 2017, doi: 10.1186/s12978-016-0268-z). On the other hand, there is plenty of research analysing the adverse consequences of unmet need (e.g., HIV, unintended pregnancies, maternal mortality). Authors fail to address this issue in the introduction as a context of non-use of contraceptives, even more when unmarried adolescents are at higher risk of having unmet need (Sánchez and Ortega 2018, doi: 10.4054/DemRes.2018.38.45).

7. Authors refer to target 3.7 of the Sustainable Development Goals but they fail to cite it explicitly.

Data

8. Authors should clarify whether they are using never married or unmarried women.

9. Authors present the total number of households of each survey and the sample they are using. However, I would like to know the proportion of the sample compared to the corresponding age group. They could include this information in both Table 2 and text.

10. Authors should include how they have defined “sexually active women”. Having had sex over the last month? Over the last year? Or are they referring to ever-had-sex women as sexually active?

11. Authors are using 2008, 2013, and 2018 DHS “…based on the understanding that a ten-year period is a reasonably long period of time”. By the same logic, a thirty-year period is even more reasonable long period and should be preferred. Why are not they using the 1990 and 2003 DHS? Of course, 1990 DHS does not include men sample; however, they could use 1990-2018 for women and 2003-2018. It is not clear to me why they are not using all available surveys.

12. Religion is included in Table 2 but then left out of the analysis. Why? It would be interesting to observe the influence of religion on contraceptive use behavior.

13. Since secondary/higher education includes roughly 90% of the sample, perhaps the addition of a third category could help show differences by educational level.

Results

14. Table 3 should specify that figures are in percentages.

15. I think that keeping the column “At least one sexual risk” in Table 4, as it is in Table 3, contributes to the analysis. I would recommend to keep the column. The same for Table 5. The multivariate analysis could also be done for “At least one sexual risk”.

16. When analyzing wealth quintiles, I would recommend to use the middle quintile as the base category. Perhaps this will make the differences between quintiles clearer.

17. I would recommend to keep the same order of columns in Table 5 as it is in Tables 3 and 4. It would help to readability.

18. Regarding the results of the logit model, analysis of non-use of modern contraceptives should be expanded.

Discussion

19. I have concerns about the conclusions drawn from the variable "sexual relations before age 18" when comparing the cohorts, as they could be misleading as they are presented now. First, except for those aged 19, all of women of ages 15-19 have had sexual intercourse before age 18 and that is reflected on the estimates of table 5. In other words, the probability of having had sex before the age of 18 for people aged 15 to 19 is practically 1 (so, no surprises there).

Second, adolescents aged 15 to 17 still have several years ahead of them to have their first sexual encounter (taking into account the imposed threshold of 18 years of age). In other words, they have not had the opportunity to complete the exposure period for which the analysis is intended. In this regard, this is why the DHS program does not publish estimates of first sexual intercourse by exact age 18 for those aged 15-19, but it does in the case of by exact age 15.

Authors should find a better (and accurate) way of presenting their results.

20. Authors do not refer at any point about the underlying causes of non-use of contraceptives and inconsistent condom use. As it is now focused, it would be understood that not using modern contraceptives is the result of a negligent decision. However, this is not usually the cases, especially among young people.

It makes me wonder, do women want to use contraceptives? Perhaps they don’t use them because they don’t want them (preferences that could result in high fertility). Or maybe they do not use them because they are not able to get them (unmet need; thus, fail in meeting SRH goals). Depending on the reasons, the policy would be designed.

From there, it would be nice to have some context from other sub-Saharan African countries. From there, authors could compare their result to those found in other countries in the region (e.g., Sedgh and Hussain 2014, doi: 10.1111/j.1728-4465.2014.00382.x; Moreira et al 2019, doi: 10.1186/s12978-019-0805-7).

21. In the cases of unmet need, not using contraceptives usually leads to unintended pregnancy. In adolescence, it is often considered one of the most common burdens of life, responsible for adverse outcomes in adulthood, like reduced labor income, lower educational achievements, or unsafe abortion (e.g., Hindin et al. 2016, doi: 10.2471/BLT.16.170688; Santhya and Jejeebhoy 2015, doi: 10.1080/17441692.2014.986169). Authors fail to discuss the consequences of unintended pregnancy due to non-use of contraceptives, which becomes more relevant in the high-fertility context of Nigeria and in the light of meeting SDG goals.

On the other hand, from a SRH rights perspective, authors should discuss about the importance that adolescents meet their contraceptive needs (e.g., Sánchez and Ortega 2018, doi: 10.4054/DemRes.2018.38.45).

22. There has been little discussion about gender disparities in risky sexual behavior. Results show, first, that women face higher risky behavior than men and, second, it has increased over time. It could give lights about gender violence, especially in recent years (from results in 2018).

23. Statement from line 288 to line 290 contradicts the results presented earlier.

24. I do not agree with the data limitations discussed by the authors:

24.1 Temporality issue: this can be controlled in the regressions.

24.2 Lack of inference of causality: this is not clear to me. Authors are not looking for causality, why should it be a problem?

24.3 Inability to explore other relevant variables outside the data collection framework of the original study: What variables do the authors refer to?

24.4 Recall bias: Authors are using questions reporting on current information and, in some cases, they go back up to 12 months before the survey. Precisely, recall bias is significantly reduced when using the most current information.

I’m not saying that there are no data limitations. I’m only pointing out that I do not agree with those included in the manuscript.

24.5 Finally, in the Introduction authors say SHR policies in Nigeria over the last two decades have been ineffective. I would recommend to discuss on potential solutions.

6. PLOS authors have the option to publish the peer review history of their article (what does this mean?). If published, this will include your full peer review and any attached files.

Reviewer #1: **Yes: **Obasanjo Bolarinwa

Reviewer #2: No

---

## [Author Response · Author response to Decision Letter 0]

25 Nov 2020

Response to reviewers’ comments

Editor’s comments

Both reviewers make interesting points regarding the analysis and interpretation of results, which technically is sound. Look, in particular, at comments by reviewer 2, and please adjuts the text accordingly.

Expanding on reviewer 2 comments, there is a change in the analysis that I see necessary. Reviewer 2 points to the difference between males and females, and looking at table 4 we see that behaviour of males and females is radically different. However, the analysis is carried out jointly for both sexes with sex only as a control variable. I don't see this as appropriate. You should fit a different equation for each sex, or, if you prefer so, use a model with full interactions among sex and the rest of variables. That would allow you to test equality of the different coefficients according to sex. It is very difficult to interpret the results from the joint analysis and I don't see a priori any reasons why the effects of covariates should be the same for both sexes when levels are so different. As you know, one of the requirement for publication in Plos One is a high technical standard and this requirement goes in that direction.

One other minor comment: Table 4 does not specify if N is weighted or unweighted. Also there is no information on the kind of weighting applied. It should be normalized weights (average 1).

Response

As advised, we have done separate analysis for both male and female respondents. We also indicated that weighting was applied in all the analysis. 

Comments

Reviewer #1: 

The authors have contributed to an important aspect of sexual and reproductive health (SRH) by looking at the prevailing factors contributing to poor SRH outcomes. The novelty of this work is in no doubt considering the conceptualization, methodology and analysis. However, the author would have employed multi-level analysis; this would have shown the degree of variation within each community of the young people; also, this would have shown other factors that contribute to poor SRH outcomes among young Nigeria as such developing required interventions to address the study conclusion will be targeted to the needed group(s) at different levels, as against socio-demographic characteristics of the respondents alone. Though the technicality of this work is sound, this could be a very good suggestion to employ in the future.

Response

Thank you. The main objective of the manuscript was to tease out the recent changes in contraceptive and sexual behaviours of young people and the analysis was done to address this objective. As pointed out by the reviewer, the advice to consider multilevel analysis for future endeavour is in order as the current study does not seek to investigate the influences of contextual variables.

Comments

Please kindly check on the manuscript submitted, from lines 56 to 58 about Nigeria being the country with the highest-burden of HIV - You might want to check the literature to be sure.

Response

We did not state that Nigeria is the “country with the highest-burden of HIV” as pointed out; rather, as shown below, what we stated in the manuscript is exactly the reality as per the most recent literature that:

“Among others, Nigeria has one of the highest burdens of Human Immunodeficiency Virus (HIV) in the world and young people aged 15-24 years are disproportionately affected by the infection, and contribute about a third of all the country’s new HIV infections.”

Comments

Also, please kindly check on the manuscript from lines 77 to 81 statement. It appears vague and not passing any message.

Response

We have revised the sentences in lines 77 to 81 accordingly. The revised texts are provided below:

“A review of SRH behaviour among young people in sub-Saharan Africa reflects an interesting contrast between males and females [1]. While higher economic status was a protective factor against early sexual initiation among young women, a reverse situation was the case among young men. Also, females mostly reported health care workers as their main source of contraceptive information while males mainly reported the media, friends and internet [2].” 

Reviewer #2: 

The manuscript is interesting and has presented the main idea clearly. The subject of this document is relevant and is within the interests of several international organizations that aim to reduce the risks associated to sexual risk behavior in order to achieve SRH-focused objectives. I think the document is on the right track, but it needs major revisions especially on the policy implications. I include first a couple of comments at the general level and then some more specific ones.

Comments

General comments

1. Authors focus their research on the interest on SRH, as it is shown in the Introduction. However, the Discussion has lack of analysis regarding the policy implications and the agenda for policymakers. I expand this comment below.

Response

We have now presented the policy implications of the study in the conclusion section. This is provided below:

“The present study lends credence to the previous findings and confirms Nigeria as a high-burden country for adolescent and youth health problems with SRH being a significant contributor. We found that prevalence of risky sexual behaviour was higher among young women than men, thus reflecting a gender-based dimension in the poor SRH outcomes. Our findings have important policy implications. There is need for gender-sensitive policies and programmes that account for specific peculiarities regarding the SRH outcomes between males and females. This study suggests that many previous policies and intervention efforts targeting adolescent and youth SRH have been largely ineffective, and therefore recommends the need to prioritise future actions that are well-tailored towards the specific needs of young men and women.”

Comments

2. Early sex is defined as having had first sexual intercourse at or before age 18. This is not necessarily the case in this paper. Maybe the authors could make it explicit and show the proportion of the sample falling into this definition. Beyond that, to generalize, author should avoid the term “early sex” and use “sexual debut during adolescence” or “sexual debut before age 18” when referring to this SRH dimension.

Response

The proportions of respondents that had first sex before age 18 have been presented in Tables 3, 4 and 5 in the manuscript. On the other comments, based on our operationalization, without any ambiguity, we clearly defined respondents who had the first sexual intercourse before age 18 in this study as having early sex. 

Comments

3. Nigeria is one of the most populated countries in the world and also one of the highest TFR. Even more, Nigeria is still on the early stage of fertility transition. Besides the potential risk of STIs due to the erratic use of condoms, non-use of modern contraceptives contributes to high levels of fertility. There is research discussing on the benefits of population reduction. Fertility preferences and desires could shape contraceptive use behavior. Authors should account for the high-fertility context of Nigeria at the moment of drawing their conclusions.

Response

We do agree that erratic use of condoms and non-use of modern contraceptives among adolescents contribute largely to high adolescent childbearing in Nigeria and this in turn has serious implications for Nigeria’s high fertility level. Thus, the revised discussion and conclusion now reflect on the negative implications of these scenarios.

Comments

4. Author use the terms unmarried and never married as synonyms, although there is a subtle difference. Unmarried includes 4 categories: never married, widowed, not living together, and divorced. It is true that among women aged 15-24 never married and unmarried are quite similar; however, authors should note this difference for further research.

Response

We have noted the subtle difference between the two terms – unmarried and never married. For consistency sake, we have used just one term (never married) throughout the manuscript. 

Comments

5. Minor language revision and proof-reading before resubmitting.

Response

A thorough language revision of the manuscript has been carried out.

Comments

Specific comments

Introduction

6. From SRH perspective, one of the biggest concerns is unmet need for contraceptives. On the one hand, causes of unmet need are highly correlated to SRH rights violations (barriers to use, gender violence, and social pressure, among others) (e.g., Machiyama et al 2017, doi: 10.1186/s12978-016-0268-z). On the other hand, there is plenty of research analysing the adverse consequences of unmet need (e.g., HIV, unintended pregnancies, maternal mortality). Authors fail to address this issue in the introduction as a context of non-use of contraceptives, even more when unmarried adolescents are at higher risk of having unmet need (Sánchez and Ortega 2018, doi: 10.4054/DemRes.2018.38.45).

Response

The revised manuscript has now addressed the context of non-use of contraceptives as shown below:

Scholars have adduced that unmarried adolescents are at higher risk of having unmet need for contraception than the older population due to social pressure, contraceptive access barriers arising from providers’ bias, and gender-based violence among others [17-19]. There are adverse consequences of unmet need among young people, such as unintended pregnancies, maternal mortality, and sexually transmitted infections, including HIV/AIDS.

Comments

7. Authors refer to target 3.7 of the Sustainable Development Goals but they fail to cite it explicitly.

Response

The target 3.7 of the Sustainable Development Goals has been cited in the revised manuscript, as presented below:

“Target 3.7 of the Sustainable Development Goals is to ensure universal access to sexual and reproductive health-care services, including for family planning, information and education, and the integration of reproductive health into national strategies and programmes by 2030”.

Comments

Data

8. Authors should clarify whether they are using never married or unmarried women.

Response

Our analysis focused on never married young men and women. We have indicated this information in the revised manuscript. 

Comments

9. Authors present the total number of households of each survey and the sample they are using. However, I would like to know the proportion of the sample compared to the corresponding age group. They could include this information in both Table 2 and text.

Response

The proportions of the sample compared to the corresponding age groups from the overall survey data have been included in Table 2 and the result section of the manuscript as follows:

“The total respondents in age group 15-19 account for 18% of the overall respondents in the age group in 2008 survey, 14% in 2013 and 11% in 2018 while respondents aged 20-24 years in this study account for 30% in 2008, 27% in 2013 and 21% in 2018.”

Comments

10. Authors should include how they have defined “sexually active women”. Having had sex over the last month? Over the last year? Or are they referring to ever-had-sex women as sexually active?

Response

We have included in the manuscript how we defined “sexually active women” which is as follows:

“This study focused on the never-married but sexually active young men and women aged 15 - 24. For the purpose of this study, the sexually active were the young men and women who reported to have ever been engaged in sexual intercourse. The purpose of this operational definition was to identify the maximum number of young men and women who had experienced sexual encounter.”

Response

Comments

11. Authors are using 2008, 2013, and 2018 DHS “…based on the understanding that a ten-year period is a reasonably long period of time”. By the same logic, a thirty-year period is even more reasonable long period and should be preferred. Why are not they using the 1990 and 2003 DHS? Of course, 1990 DHS does not include men sample; however, they could use 1990-2018 for women and 2003-2018. It is not clear to me why they are not using all available surveys.

Response

A number of studies have analysed SRH behaviour in the distant past using the 1990 and 2003 data. Besides, the 1990 data is reputed to be of lower quality with respect to many of the key indicators measured in the survey. We chose to focus on 2013-2018 data which have better data quality. Besides, our choice of 2008 – 2018 will serve as update to the existing study with the aim of teasing out what has changed in the more recent time.

Comments

12. Religion is included in Table 2 but then left out of the analysis. Why? It would be interesting to observe the influence of religion on contraceptive use behavior.

Response

We have included religion in the analysis

Comments

13. Since secondary/higher education includes roughly 90% of the sample, perhaps the addition of a third category could help show differences by educational level.

Response

We initially categorised education as (i) no/Primary education and (ii) Secondary/higher education due to the very small proportion of the young people in no education and higher education groups. However, in line with the above comments, the education variable has been re-categorized as: (i) No education (ii) primary education (iii) secondary education (iv) tertiary education

Comments

Results

14. Table 3 should specify that figures are in percentages.

15. I think that keeping the column “At least one sexual risk” in Table 4, as it is in Table 3, contributes to the analysis. I would recommend to keep the column. The same for Table 5. The multivariate analysis could also be done for “At least one sexual risk”.

16. When analyzing wealth quintiles, I would recommend to use the middle quintile as the base category. Perhaps this will make the differences between quintiles clearer.

17. I would recommend to keep the same order of columns in Table 5 as it is in Tables 3 and 4. It would help to readability.

18. Regarding the results of the logit model, analysis of non-use of modern contraceptives should be expanded.

Responses

We have shown in the footnote of Table 3 that figures are in percentages. The column “At least one sexual risk” has been included in Tables 4 and 5 as advised, and the results have been interpreted in the result section of the manuscript. We understand the point of view of the reviewer with respect to the selection of “middle” wealth quintile as the base category. We have selected the middle wealth quintile as the reference category as advised. The order of Tables 5 has been modified to follow the same order in Tables 3 and 4.

Comments

Discussion

19. Authors do not refer at any point about the underlying causes of non-use of contraceptives and inconsistent condom use. As it is now focused, it would be understood that not using modern contraceptives is the result of a negligent decision. However, this is not usually the cases, especially among young people. It makes me wonder, do women want to use contraceptives? Perhaps they don’t use them because they don’t want them (preferences that could result in high fertility). Or maybe they do not use them because they are not able to get them (unmet need; thus, fail in meeting SRH goals). Depending on the reasons, the policy would be designed. From there, it would be nice to have some context from other sub-Saharan African countries. From there, authors could compare their result to those found in other countries in the region (e.g., Sedgh and Hussain 2014, doi: 10.1111/j.1728-4465.2014.00382.x; Moreira et al 2019, doi: 10.1186/s12978-019-0805-7).

Response 

In response to these comments, we have included the information below in the revised manuscript:

As previously established in other SSA countries [19, 31, 32], non-use of contraceptives and erratic condom use among young people are perhaps due to unmet need for contraception, lack of youth friendly services, peer-pressure and personal preferences. These factors have implication and potential for sustaining high population growth in Nigeria which has a high-fertility context. Moreover, scholars have adduced that unintended pregnancy in adolescence is a major cause of adverse social and health outcomes, including unsafe abortion, lower educational achievements, low employment opportunities and reduced labour income in adulthood[33].

Comments

21. In the cases of unmet need, not using contraceptives usually leads to unintended pregnancy. In adolescence, it is often considered one of the most common burdens of life, responsible for adverse outcomes in adulthood, like reduced labor income, lower educational achievements, or unsafe abortion (e.g., Hindin et al. 2016, doi: 10.2471/BLT.16.170688; Santhya and Jejeebhoy 2015, doi: 10.1080/17441692.2014.986169). Authors fail to discuss the consequences of unintended pregnancy due to non-use of contraceptives, which becomes more relevant in the high-fertility context of Nigeria and in the light of meeting SDG goals. On the other hand, from a SRH rights perspective, authors should discuss about the importance that adolescents meet their contraceptive needs (e.g., Sánchez and Ortega 2018, doi: 10.4054/DemRes.2018.38.45).

Response 

These comments have been addressed by including relevant additional information in the discussion section. 

22. There has been little discussion about gender disparities in risky sexual behavior. Results show, first, that women face higher risky behavior than men and, second, it has increased over time. It could give lights about gender violence, especially in recent years (from results in 2018).

Response 

We have now discussed the issue of gender disparities in risky sexual behaviour between young men and women in the discussion section. 

Our results show that young women face higher risky sexual behaviour than men, and this disparity has increased over time. This perhaps reflects increase in sexual and gender violence in the recent years.

Comments

23. Statement from line 288 to line 290 contradicts the results presented earlier.

Response 

Accordingly, the statement has been revised for clarity, as presented below:

“Broadly, our results showed a higher level of risky sexual behaviour among the respondents in 2018 compared to 2008”

Comments

24. I do not agree with the data limitations discussed by the authors:

24.1 Temporality issue: this can be controlled in the regressions.

24.2 Lack of inference of causality: this is not clear to me. Authors are not looking for causality, why should it be a problem?

24.3 Inability to explore other relevant variables outside the data collection framework of the original study: What variables do the authors refer to?

24.4 Recall bias: Authors are using questions reporting on current information and, in some cases, they go back up to 12 months before the survey. Precisely, recall bias is significantly reduced when using the most current information.

I’m not saying that there are no data limitations. I’m only pointing out that I do not agree with those included in the manuscript.

Response 

We have expunged the limitations recommended for deletion, and have included the key limitations of the study.

Comments

24.5 Finally, in the Introduction authors say SHR policies in Nigeria over the last two decades have been ineffective. I would recommend to discuss on potential solutions.

Response 

We have made suggestions in the conclusion section on how to make future SRH policies in Nigeria more effective than the previous efforts. The suggested solution is presented below:

“We found that prevalence of risky sexual behaviour was higher among young women than men, thus reflecting a gender-based dimension in the poor SRH outcomes. Our findings have important policy implications. There is need for gender- specific policies and programmes that account for specific gender peculiarities regarding the SRH outcomes between males and females. This study suggests that many previous policies and intervention efforts targeting adolescent and youth SRH have been largely ineffective, and therefore recommends the need to prioritise future actions well-tailored towards the specific needs of young men and women.”

---

## [Decision Letter · Decision Letter 1]

7 Dec 2020

PONE-D-20-27905R1

Changes in contraceptive and sexual behaviours among unmarried young people in Nigeria: evidence from nationally representative surveys

PLOS ONE

Dear Dr. Adedini,

Thank you for submitting your manuscript to PLOS ONE. After careful consideration, we feel that it has merit but does not fully meet PLOS ONE’s publication criteria as it currently stands. Therefore, we invite you to submit a revised version of the manuscript that addresses the points raised during the review process.

Reviewer 1, who proposed some changes is satisfied. Reviewer 2 has not been invited since they were already willing to accept.  The manuscript has improved to a large extent, and the main concern regards the lack of visibility of the who is unmarried sexually active. These are the minor changes suggested:

Religion has been added to the analysis but not to table 1.Avoid the use of the word “respondents” in discussing table 2, since you are not talking about the characteristics of the survey respondents: the biased characteristics depend on the selection strategy:  never married sexually active. That has to be born in mind in the interpretation of results, which currently is not (lines 174), which should be rewritten after redesign as suggested below.As reviewer 1 originally pointed out, this is important for interpretation. The current design of table 2 incorporating the proportion only for the age-group with a different format and a footnote is not satisfactory. Reading the table one wonders why there is an overrepresentation of wealthier / more educated. It could be because of sexual activity or because of selection in marriage, and those go in different directions. Please redesign the table in the following way:Omit absolute numbers, except in the header (as it currently stands)Provide nine columns, three for each survey, including percentages by socidemographic characteristics for the three subgroups of married, not sexually active, and unmarried sexually active (your group of interest). You could include also a chi-square test of association if desired.The interpretation for the first part of results becomes then much more focused: most likely the over representation of wealthier/more educated will be due to lower prevalence of marriage, but there might also be differences in sexual activity. Just rephrase. Line 118: Remove “been” should be “to have ever engaged in sexual intercourse”Line 445, 449 highlight higher risk for female, but this is not true for all dimensions (multiple partners goes in the different direction). Please rephrase so that it leads naturally to lines 451-452 saying that SRH policies need to “ account for specific gender peculiarities”.Rephrase line 80 as “and pf the adverse consequences of unmet need among young people such as”

We look forward to receiving your revised manuscript.

Kind regards,

José Antonio Ortega, Ph.D.

Academic Editor

PLOS ONE

Reviewers' comments:

Reviewer's Responses to Questions

**Comments to the Author**

1. If the authors have adequately addressed your comments raised in a previous round of review and you feel that this manuscript is now acceptable for publication, you may indicate that here to bypass the “Comments to the Author” section, enter your conflict of interest statement in the “Confidential to Editor” section, and submit your "Accept" recommendation.

Reviewer #2: All comments have been addressed

2. Is the manuscript technically sound, and do the data support the conclusions?

Reviewer #2: (No Response)

3. Has the statistical analysis been performed appropriately and rigorously? 

Reviewer #2: (No Response)

4. Have the authors made all data underlying the findings in their manuscript fully available?

Reviewer #2: (No Response)

5. Is the manuscript presented in an intelligible fashion and written in standard English?

Reviewer #2: (No Response)

6. Review Comments to the Author

Reviewer #2: (No Response)

7. PLOS authors have the option to publish the peer review history of their article (what does this mean?). If published, this will include your full peer review and any attached files.

Reviewer #2: No

---

## [Author Response · Author response to Decision Letter 1]

6 Jan 2021

Step-by-step responses to the comments

Comments

Religion has been added to the analysis but not to table 1.

Response

We have included religion in Table 1 with the corresponding operational definition (Religious affiliation of the respondents) and grouping for analytical purpose ((1) Christianity; (2) Islam; (3) Traditional and others)

Comment

Avoid the use of the word “respondents” in discussing table 2, since you are not talking about the characteristics of the survey respondents: the biased characteristics depend on the selection strategy: never married sexually active. That has to be born in mind in the interpretation of results, which currently is not (lines 174), which should be rewritten after redesign as suggested below.

• As reviewer 1 originally pointed out, this is important for interpretation. The current design of table 2 incorporating the proportion only for the age-group with a different format and a footnote is not satisfactory. Reading the table one wonders why there is an overrepresentation of wealthier / more educated. It could be because of sexual activity or because of selection in marriage, and those go in different directions. Please redesign the table in the following way:

o Omit absolute numbers, except in the header (as it currently stands)

o Provide nine columns, three for each survey, including percentages by socidemographic characteristics for the three subgroups of married, not sexually active, and unmarried sexually active (your group of interest). You could include also a chi-square test of association if desired.

o The interpretation for the first part of results becomes then much more focused: most likely the over representation of wealthier/more educated will be due to lower prevalence of marriage, but there might also be differences in sexual activity. Just rephrase.

Response

The result in Table 2 has been revised and interpreted as advised

Comments

Line 118: Remove “been” should be “to have ever engaged in sexual intercourse”

Response

The word “been” in Line 118 has been removed from the sentence which is now revised as “to have ever engaged in sexual intercourse”

Comments

Line 445, 449 highlight higher risk for female, but this is not true for all dimensions (multiple partners goes in the different direction). Please rephrase so that it leads naturally to lines 451-452 saying that SRH policies need to “account for specific gender peculiarities”.

Response

The sentences in Line 445 and 449 have been rephrased and expanded.

Comments

Rephrase line 80 as “and of the adverse consequences of unmet need among young people such as”

Response

The sentence in line 80 has been rephrased as suggested by the reviewer

---

## [Editor Report · Decision Letter 2]

18 Jan 2021

Changes in contraceptive and sexual behaviours among unmarried young people in Nigeria: evidence from nationally representative surveys

PONE-D-20-27905R2

Dear Dr. Adedini,

We’re pleased to inform you that your manuscript has been judged scientifically suitable for publication and will be formally accepted for publication once it meets all outstanding technical requirements.

One minor detail that has to be addressed is, in table 2, writing "Ever married" where currently stated "Married". This is in line with the comment made in the original draft by a reviewer suggesting clarification of whether unmarried meant unmarried or never married. I assume it really is "Ever married".

Kind regards,

José Antonio Ortega, Ph.D.

Academic Editor

PLOS ONE
---

## [Editor Report · Acceptance letter]

22 Jan 2021

PONE-D-20-27905R2 

Changes in contraceptive and sexual behaviours among unmarried young people in Nigeria: evidence from nationally representative surveys 

Dear Dr. Adedini:

I'm pleased to inform you that your manuscript has been deemed suitable for publication in PLOS ONE. Congratulations! Your manuscript is now with our production department. 

Kind regards, 

on behalf of

Dr. José Antonio Ortega 

Academic Editor

PLOS ONE